# Determining the interlayer shearing in twisted bilayer MoS$_2$ by nanoindentation

Yufei Sun[1,4], Yujia Wang [2,4], Enze Wang[1,4], Bolun Wang[1,4], Hengyi Zhao[2], Yongpan Zeng[2], Qinghua Zhang[3], Yonghuang Wu[1], Lin Gu [3], Xiaoyan Li [2✉] & Kai Liu [1✉]

The rise of twistronics has increased the attention of the community to the twist-angle-dependent properties of two-dimensional van der Waals integrated architectures. Clarification of the relationship between twist angles and interlayer mechanical interactions is important in benefiting the design of two-dimensional twisted structures. However, current mechanical methods have critical limitations in quantitatively probing the twist-angle dependence of two-dimensional interlayer interactions in monolayer limits. Here we report a nanoindentation-based technique and a shearing-boundary model to determine the interlayer mechanical interactions of twisted bilayer MoS$_2$. Both in-plane elastic moduli and interlayer shear stress are found to be independent of the twist angle, which is attributed to the long-range interaction of intermolecular van der Waals forces that homogenously spread over the interfaces of MoS$_2$. Our work provides a universal approach to determining the interlayer shear stress and deepens the understanding of twist-angle-dependent behaviours of two-dimensional layered materials.

[1] State Key Laboratory of New Ceramics and Fine Processing & Key Laboratory of Advanced Materials of Ministry of Education, School of Materials Science and Engineering, Tsinghua University, Beijing 100084, China. [2] Center for Advanced Mechanics and Materials, Applied Mechanics Laboratory, Department of Engineering Mechanics, Tsinghua University, Beijing 100084, China. [3] Institute of Physics, Chinese Academy of Sciences, Beijing 100190, China. [4] These authors contributed equally: Yufei Sun, Yujia Wang, Enze Wang, Bolun Wang. ✉email: xiaoyanlithu@tsinghua.edu.cn; liuk@tsinghua.edu.cn

Two-dimensional (2D) layered materials have attracted considerable attention in the past decade owing to their physical and chemical properties. Their atomically flat, dangling-bond free surface enables van der Waals (vdW) stacking and the integration of 2D materials into 3D architectures, providing an additional dimension for the modulation of material properties[1–3]. The twist angle, which determines the vdW stacking direction from one layer of 2D material to another one, should influence the properties of integrated 2D materials[4–7] but was commonly ignored in the early studies of 2D electronic devices. However, a recent study has shown that a vdW-stacked bilayer graphene exhibits superconductivity at a specific twist angle of 1.1° (the first "magic" angle)[8], leading to the rise of "twistronics". In recent years, twist-angle-dependent correlated insulator states, Moiré excitons, stacking-dependent interlayer magnetism, and topological polaritons have been discovered[9–13]. Inspired by these studies, there are growing demands to understand in depth how the twist angle influences the interlayer coupling of 2D homo- or heterostructures. Although many studies have focused on electronic interlayer coupling in twistronics, the relationship between interlayer mechanical interactions and twist angles has yet to be reported.

It is of importance to clarify the relationship between the twist angle and the interlayer mechanical interaction in vdW-integrated architectures, which, in particular, benefits the design of 2D flexible electronics[14]. In 2D layered systems, the overall robustness is determined by the interlayer mechanical interaction rather than the mechanical strength of each individual layer, as the interlayer vdW forces are much weaker than the intralayer chemical bonding forces[15]. Unfortunately, current methods of measuring interlayer interactions of 2D materials, including the pressurized bubbling method[16–19], tip-based adhesion force measurement[20–22], and nanoindentation[23–28], have certain limitations when probing the twist-angle-dependent interlayer interaction of 2D materials in monolayer limits. For instance, pressurized bubbling tests require the ultimate gas impermeability of detected materials to determine their interlayer shear stresses, and thus, the detected materials are usually limited to graphene[16–18]. The tip-based adhesion force measurement could determine the adhesive force between the 2D material-wrapped tip and the target 2D material[20,21], yet this method does not have twist-angle-resolved capability. Nanoindentation has been widely used to measure the elastic moduli of 2D materials by indenting suspended regions of 2D materials. It could also qualitatively probe the interlayer interactions of bilayer or multilayer 2D materials because weaker interlayer interactions induce greater attenuation of the effective elastic moduli, which are lower than the overall moduli counting each layer[23–28]. However, it is still challenging to quantitatively determine the interlayer shear stress because the indentation induces tensile stress and shear stress simultaneously at the suspended region of bilayer or multilayer 2D materials.

In this work, we established an experimental configuration together with theoretical model to probe the twist-angle-dependent interlayer interaction of twisted bilayer MoS₂ (TBLM) by nanoindentation. Experimentally, this is realized by first selectively breaking the suspending region of the bottom layer of TBLM over circular holes on a substrate and then twistedly stacking the upper layer onto the bottom layer and keeping the upper layer intactly suspended over the holes (Fig. 1a). In this configuration, the suspended region of the TBLM is only from the upper layer, which is constrained by the bottom layer around the edges of the holes. As a result, the tensile and shear regions of the upper layer MoS₂ are separated, and the shearing/sliding interaction between the two layers only occurs at the boundaries around the edges of the holes. This experimental configuration enables us to build a clear realistic theoretical model based on shearing boundaries to describe the interlayer interaction of the TBLM. Although the nonplanar crystal structure of MoS₂ (i.e., one layer of Mo atoms sandwiched by two layers of S atoms) implies a significant steric effect[29] and thus a twist-angle-dependent interlayer mechanical interaction, our results show that the interlayer shearing interaction is surprisingly independent of the twist angle. With the shearing-boundary mechanical model, the average interlayer shear stress of the TBLM is quantitatively determined to be ~2.51 MPa. This value is much lower than that of the MoS₂@SiO₂ interface (11.09 MPa), suggesting that TBLM is more prone to interlayer shearing than monolayer MoS₂ laid on SiO₂. Molecular dynamics (MD) simulations further confirm the twist-angle-independent interlayer mechanical interaction, and the derived theoretical interlayer shear stress is very consistent with our experimental data. The independence of the interlayer shear stress of TBLM is attributed to the fact that the overall interlayer vdW force is the sum of intermolecular forces, which homogenously spread over the TBLM interfaces. Our work provides a universal approach to quantitatively evaluate the interlayer interactions of various 2D materials and their heterostructures. The twist-angle-independent shear stress also sheds light on the fabrication and application of 2D vertical heterostructures.

## Results and discussion

**Preparation and characterization of TBLM.** High-quality MoS₂ monolayers were grown on SiO₂/Si substrates (Supplementary Fig. 1) under ambient pressure using MoO₃ and sulfur as precursors with the assistance of perylene-3,4,9,10-tetracarboxylic acid tetrapotassium salt (PTAS), which is similar to previous reports[30,31]. To carry out the nanoindentation tests, a SiO₂/Si substrate was prepatterned with arrays of circular holes with a depth of 300 nm and a diameter of either ~1.0 μm or ~1.5 μm[32]. TBLM was prepared by two-step transfer processes that include a polymethyl methacrylate (PMMA)-assisted wet transfer to break the bottom MoS₂ monolayer region over holes, followed by a polydimethylsiloxane (PDMS)-assisted dry transfer to stack and suspend the upper MoS₂ monolayer region over holes (Fig. 1a, also see Methods for details). After the wet transfer, the bottom MoS₂ monolayers collapsed over holes and exhibited sharp edges around the holes, while only the supported region remained on the substrate (Supplementary Fig. 2). Then, after the dry transfer, the upper MoS₂ monolayers were randomly stacked onto the bottom monolayers, forming TBLM with random twist angles (Fig. 1b). No solvent treatment was involved when the PDMS was peeled off in the dry transfer to keep the suspended region of the upper monolayers intact. Either the wet or dry transfer process was kept clean in all aspects to guarantee clean surfaces of MoS₂ monolayers (Supplementary Fig. 1). The sample was also annealed after either transfer process to remove any polymer residues and have the twisted bilayer interact effectively. This random stacking is efficient for preparing clean TBLM with various twist angles to obtain abundant angle-resolved data. As the MoS₂ monolayers exhibit regularly triangular shapes, the twist angle of TBLM can be directly determined by identifying the crystal orientations of the upper and bottom MoS₂ monolayers under an optical microscope (Fig. 1b). After the two-step transfer, only the upper MoS₂ monolayers are suspended over the holes and constrained by the bottom MoS₂ monolayers around the edges of the holes (Fig. 1c). There only exists in the TBLM samples a very limited density of bubbles or wrinkles with a coverage reaching the lows in the twisted samples reported (Supplementary Fig. 3). Furthermore, clear Moiré patterns observed under annular dark-field scanning transmission electron

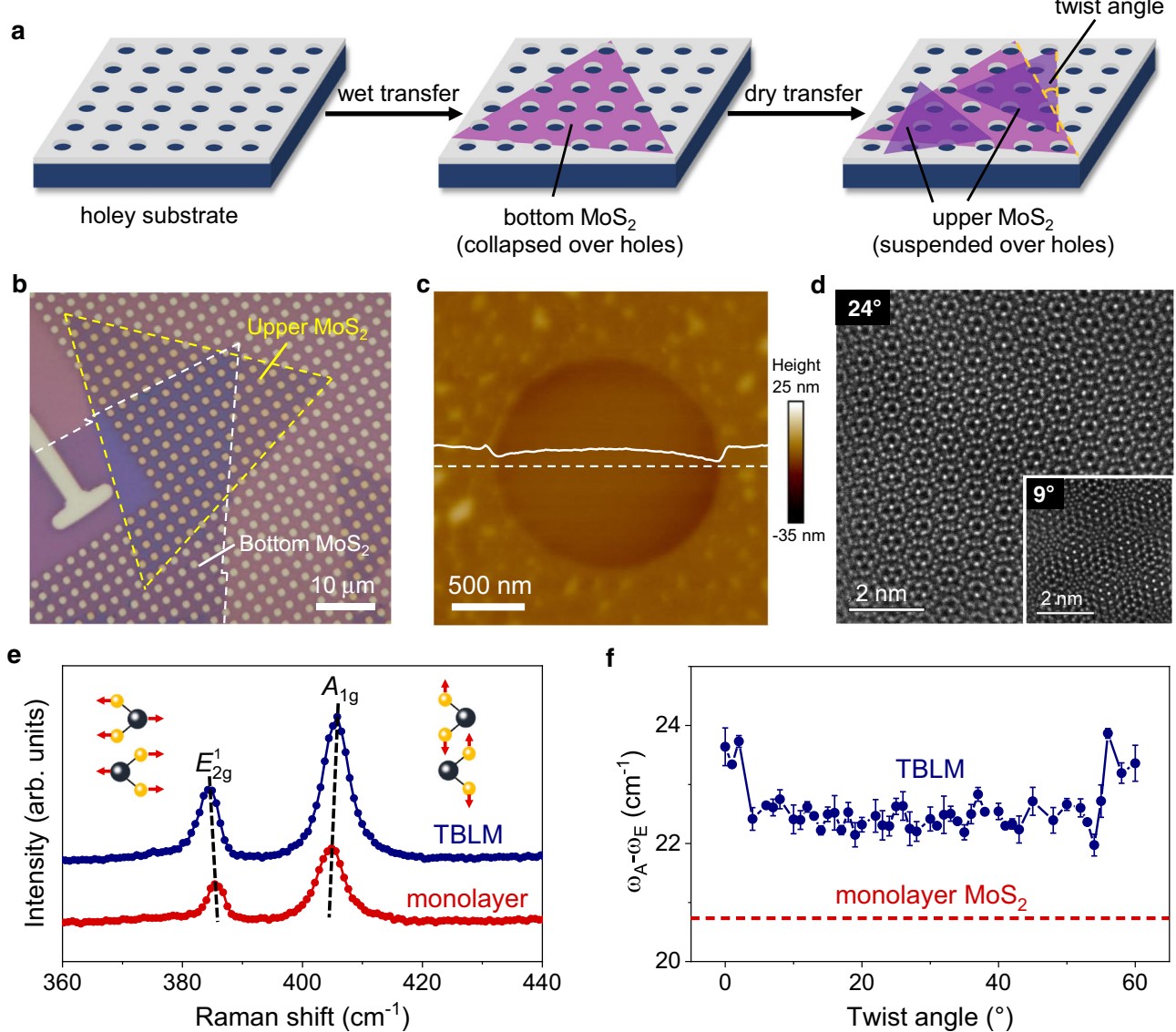

**Fig. 1 Preparation and characterization of twisted bilayer MoS$_2$ (TBLM). a** Schematic diagram of the preparation of TBLM. Here, the bottom layer MoS$_2$ collapses over the holes, while the upper layer is suspended over the holes. **b** Optical image of a TBLM sample. The dashed lines represent the edges of upper and bottom MoS$_2$ monolayers. **c** Atomic force microscope surface topology image of TBLM over a single hole. The white solid line is the height profile of the sample across the white dashed line. The variation of the height profile is 4.1 nm. **d** Clear Moiré patterns of TBLM at twist angles of 24° and 9° (inset) observed under annular dark-field scanning transmission electron microscopy. **e** Comparison of the Raman spectra of TBLM and monolayer MoS$_2$. The increase in peak interval is illustrated by the two black dashed lines. The inset shows the vibrational modes of $E^1_{2g}$ and $A_{1g}$. **f** Twist-angle dependence of the peak interval in TBLM samples on a SiO$_2$/Si substrate. The red dashed line indicates the peak interval between $E^1_{2g}$ and $A_{1g}$ of monolayer MoS$_2$.

microscopy (ADF-STEM) also suggest the high-quality and clean interfaces of the TBLM samples (Fig. 1d).

Raman spectroscopy can be used to probe the interlayer coupling of MoS$_2$ with the fingerprint out-of-plane and in-plane vibrational modes, namely, $A_{1g}$ and $E^1_{2g}$, respectively. The peak interval between $A_{1g}$ and $E^1_{2g}$ has been found to be sensitive to the number of layers and the twist angle of MoS$_2$ owing to the interlayer coupling and different symmetries[33]. Figure 1e shows that the interval between these two characteristic Raman peaks of MoS$_2$ increases by ~2 cm$^{-1}$ for TBLM compared to monolayer MoS$_2$, indicating the existence of a strong interlayer coupling of TBLM[33,34]. Figure 1f and Supplementary Fig. 4 show that the peak interval is largest (~23.5 cm$^{-1}$) when the twist angle is close to or equal to 0 and 60°, while it remains a constant value of ~22.5 cm$^{-1}$ for other twist angles. These results correspond well

with a previous study on TBLM directly grown by CVD[4], indicating that our transfer method works for the preparation of TBLM with strong interlayer coupling.

**Nanoindentation experiments of TBLM.** We conducted nanoindentation experiments under an atomic force microscope (AFM) by applying a point force $F$ to the suspended region of a sample (Fig. 2a). The force can be calculated as $F = kx$, where $k$ and $x$ are the spring constant and the displacement of the AFM probe, respectively. Here, $k$ is calibrated by the Sader method (online calibration), which follows a simple harmonic oscillation model[32], and $x$ is given by the AFM system. The indentation depth $\delta$ of the suspended membrane can be derived as $\delta = z\text{-}x$, where $z$ is the moving distance between the tip and the sample, as illustrated in Fig. 2a. Previous theoretical studies based on the

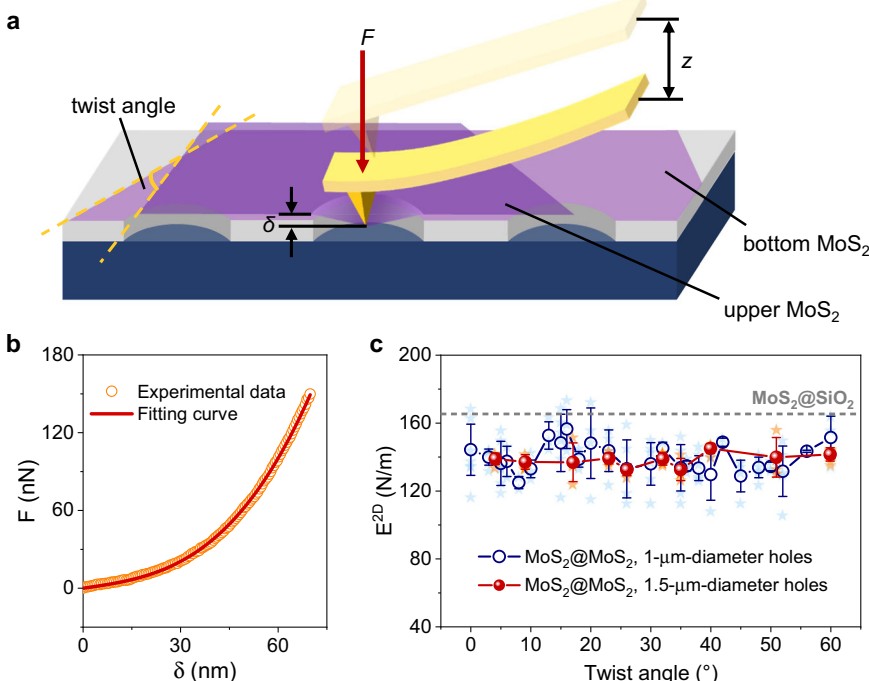

**Fig. 2 Nanoindentation experiments of TBLM. a** Schematic diagram of nanoindentation tests. *F* is the point force applied to the suspended region of the sample, *z* is the moving distance between the tip and the sample, and *δ* is the indentation depth of the suspended membrane. **b** Typical *F-δ* curve (orange circles) of a TBLM sample with the red line showing a fitting of the *F-δ* curve following Eq. (1). **c** Dependence of measured in-plane elastic moduli of TBLM on twist angles. Blue stars and orange stars mark the measured moduli of each suspended membrane over 1-μm-diameter and 1.5-μm-diameter holes, respectively. The blue circles and red spheres are the corresponding average values. Error bars represent standard deviations. The gray dashed line represents the in-plane moduli of monolayer MoS₂ on SiO₂.

fixed-boundary model, which means the suspended membrane is firmly clamped at the edge of the hole during nanoindentation, gave the following *F-δ* expression for nanoindentation of thin membranes[35–37]:

$$ F = (\sigma_0^{2D}\pi)\delta + (E^{2D}\frac{q^3}{a^2})\delta^3 \qquad (1) $$

where $E^{2D}$ is the in-plane elastic modulus in units of N/m, $\sigma_0^{2D}$ is the pretension of the suspended membrane in units of N/m, *a* is the radius of the hole, and $q = 1/(1.05-0.15\nu-0.16\nu^2)$ is a factor determined by Poisson's ratio *ν*. For MoS₂, we take $\nu = 0.27$ and $q = 1.00$, following previous studies[26,27]. Figure 2b shows a typical *F-δ* curve. By fitting the *F-δ* curve using Eq. (1), we can obtain $\sigma_0^{2D}$ and $E^{2D}$, as shown by the red line in Fig. 2b (the data processing can be seen in Supplementary Fig. 5). Equation (1) includes the asymptotic solutions at small and large displacements. For the small displacement, the one-order term related to pretension is dominant. For the large displacement, the cubic term related to the in-plane modulus is dominant. Because Eq. (1) captures the main deformation features (especially the cubic term at the large displacement) of nanoindentation and has a simple and explicit expression, it has been widely used to extract the in-plane stiffness (or in-plane modulus) of various 2D materials from nanoindentation force–displacement curves[23,25,26,28,37]. Note that Eq. (1) does not consider the influence of the indenter tip radius, and thus it introduces a certain error[38]. However, the accuracy of extracting the in-plane modulus from the nanoindentation force–displacement curve is mainly determined by the cubic term in Eq. (1). If only there are enough experimental data falling in the large displacement regime (i.e., following the cubic term), it is possible to use Eq. (1) to determine the modulus with high precision[28].

We tested 113 TBLM samples with 26 twist angles over 1-μm-diameter holes and 46 TBLM samples with 10 twist angles over 1.5-μm-diameter holes. On each sample, we performed 3–5 consecutive nanoindentations under different loads, typically ranging from 90 to 450 nN, to measure the pretensions and moduli of TBLM samples. The corresponding indentation depth is much smaller than the diameter of holes (30–70 nm for holes in 1 μm diameter and 60–130 nm for holes in 1.5 μm diameter) under these moderate loads, and as a result, the strain applied on the upper MoS₂ monolayer is estimated to be less than 2% for all of the nanoindentation measurements. Under such small strains, both the deformation of the suspended upper monolayer region and the shearing at the twisted bilayer region are elastic rather than plastic, and the in-plane deformation of the upper monolayer is very minor compared with the twist-angle-induced lattice mismatch. The *F-δ* curves in five consecutive nanoindentations under different loads follow nearly identical traces until the breaking of the upper MoS₂ monolayers (Supplementary Fig. 6), and the measured $E^{2D}$ does not change with time (Supplementary Fig. 7, $E^{2D}$ varies <5% in 2 months), suggesting very good reproducibility of our measurements. The unchanged surface topology of a TBLM sample reveals no wrinkling before and after nanoindentation (Supplementary Fig. 8). This fact excludes the wrinkling effect[39] that may be induced by nanoindentation and simplifies our model, as will be discussed later.

By fitting the force curves with Eq. (1), we obtained the $E^{2D}$ of each nanoindentation and averaged the values for each sample. Figure 2c shows the dependence of $E^{2D}$ on the twist angles. The data dispersion originates from many aspects in our experiments, such as the difference between single-crystal flakes, the offset of indentation positions, and the deviation of measured hole sizes. Note that the data obtained from the samples over larger holes

(~1.5 μm in diameter) have smaller deviations. Considering these deviations in the force measurements, the in-plane moduli of $MoS_2$ seem to remain relatively constant regardless of the twist angles (also see Supplementary Fig. 9), either for the samples over 1-μm-diameter holes or for those over 1.5-μm-diameter holes. This is a surprising result because for a $MoS_2$ monolayer, Mo atoms are sandwiched between two layers of S atoms, forming a nonplanar structure[29], which implies that $MoS_2$ should exhibit a significant steric effect and that the interlayer mechanical interaction is likely to depend on the twist angle.

There are two contradictory hypotheses that can be put forward to explain our experimental results. One is that the interlayer interaction may be dependent on the twist angle, but the interaction is strong enough to have the upper $MoS_2$ monolayer fulfill the fixed-boundary condition, and thus all the measured moduli should be equal to the intrinsic value of $MoS_2$ regardless of the twist angle. The other hypothesis is that the weak interlayer mechanical interaction has already softened the upper $MoS_2$ monolayer, but it is independent of the twist angle, so its impact on the measured moduli is identical. To clarify this, the boundary conditions of the suspended upper monolayer must be examined in depth.

**Investigation of the fixed-boundary condition model**. The current fixed-boundary mechanical model for the nanoindentation test is based on the premise that the sample is firmly clamped at the edge of the hole during nanoindentation, and the measured moduli should equal the intrinsic value (180 N/m according to the literature)[27]. This premise is widely applied to $SiO_2$/Si substrates, as previous studies have suggested a strong mechanical interaction between 2D materials and $SiO_2$ surfaces[16–18]. However, in our experiments, this premise is challenged because the interface is $MoS_2$–$MoS_2$ instead of $MoS_2$–$SiO_2$. To test this hypothesis, we also conducted nanoindentation measurements on the suspended $MoS_2@SiO_2$ region of the same flake. Figure 3a shows the typical $F$-$\delta$ curves in logarithmic coordinates for $MoS_2@MoS_2$ and $MoS_2@SiO_2$ samples. The dashed lines (with the slopes of 1 and 3 in logarithmic coordinates) are plotted as indications that at small $\delta$, $F$ increases linearly with $\delta$, which is dominated by $\sigma_0{}^{2D}$, while at larger $\delta$, $F$-$\delta$ has a cubic relationship dominated by $E^{2D}$. In this figure, the $F$-$\delta$ curve of the $MoS_2@SiO_2$ sample lies under that of the $MoS_2@MoS_2$ sample at first, while it surpasses the latter at larger $\delta$. This result indicates that $MoS_2@SiO_2$ exhibits a larger $E^{2D}$ with a smaller $\sigma_0{}^{2D}$ for this group of samples. We plotted the histograms of all measured moduli and pretension data, both well following the Gaussian distribution, as shown in Fig. 3b, c. The statistical average $E^{2D}$ measured at a specific twist angle ranges from 125–153 N/m and 132–144 N/m for the $MoS_2@MoS_2$ samples over 1.0-μm-diameter holes and 1.5-μm-diameter holes, respectively, apparently lower than the measured modulus of $MoS_2@SiO_2$ (165 N/m) (Fig. 2c). This result suggests that the interface interaction and the boundary conditions of $MoS_2@MoS_2$ and $MoS_2@SiO_2$ should be different.

**Establishment of the shearing-boundary model**. Considering that the Young's modulus is an intrinsic property of a material, independent of loading and boundary conditions, the moduli of 2D materials should be constant regardless of whether the nanoindentation tests of $MoS_2$ are performed on $SiO_2$ or $MoS_2$ substrates. The difference between the $F$-$\delta$ curves on different substrates is attributed to different interfacial sliding between the tested $MoS_2$ layer and the $SiO_2$ or $MoS_2$ substrate. However, in Eq. (1), it is assumed that the tested membrane is clamped at the edge of the hole during nanoindentation, which means an infinite

shear stress between the tested membrane and the substrate. It is obvious that such an assumption is not as realistic as the real experiments[40,41]. If using Eq. (1) to characterize/fit the nanoindentation curves of the same membrane on different substrates, then one might obtain different moduli when the interfacial interactions between membranes and substrates are distinct. To ensure the consistency of moduli measured from two different substrates, we developed a realistic theoretical model by considering a finite interfacial shear stress between the tested membrane and the substrate (see Fig. 3d). For simplicity, the shear stress is assumed to be constant and distributed in an annular shear zone. Such a model can be applicable to the nanoindentation of thin membranes (even ultrathin 2D materials) on any substrates. Based on this model, we derived the following analytical force–displacement relationship for an indented membrane on a given substrate:

$$F = (\sigma_0 h\pi)\delta + (Eh\frac{q^3}{a^2})\delta^3 + \frac{1}{2}(1+\nu)\pi\tau a^2 \left(\frac{\delta}{a}\right)\left[-1 - \frac{\sigma_0 h}{\tau a} - \frac{C_\pi Ehq^2}{4\tau a}\left(\frac{\delta}{a}\right)^2\right.$$
$$\left. + \left(1 + \frac{3\sigma_0 h}{\tau a} + \frac{3C_\pi Ehq^2}{4\tau a}\left(\frac{\delta}{a}\right)^2\right)^{\frac{1}{3}}\right]$$

(2)

where $E$ is the in-plane elastic modulus in units of $N/m^2$, $\sigma_0$ is the pretension of the suspended membrane in units of $N/m^2$, $h$ is the membrane thickness, $C_\pi = \left(\frac{3}{\pi}\right)^{\frac{2}{3}}$ is a constant and $\tau$ is the interfacial shear stress. In comparison to Eq. (1), the third term is the correction related to interfacial shear stress. Note that when expanding Eq. (2) via the Taylor series approximation and considering an infinite shear stress limitation ($\tau \to \infty$), the third term of Eq. (2) will be zero so that this equation is transformed to Eq. (1), which is consistent with our prediction. More details about the derivations of the theoretical model and Eq. (2) are given in the Supplementary Information.

We fitted the experimental force curves using the least-squares method to approach the actual value of interlayer shear stress. When we use Eq. (2) to fit the experimental curves, the modulus of $MoS_2$ is fixed and taken as the measured average value from nanoindentation for $MoS_2@SiO_2$ since the modulus is a material constant. Here, we mainly extract the interlayer shear stress by using Eq. (2) to fit the experimental curve. Undoubtedly, one can extract all three parameters (including modulus, interlayer shear stress, and pretension) via the nonlinear fitting method. Figure 3e shows the comparisons between one typical experimental curve and fitted curves based on Eq. (2). When $\tau = 2.795$ MPa, the fitted curve nearly coincides with the experimental curve, as evidenced by the minimum fitting error shown in the inset of Fig. 3f. We fitted all 424 nanoindentation measurement curves of the $MoS_2$ monolayer with different twist angles with respect to $MoS_2@MoS_2$; the obtained values of interfacial shear stress $\tau$ are summarized in Fig. 3f, with a statistical average value of 2.51 MPa. We used Eq. (2) to further fit the nanoindentation measurement curves of $MoS_2@SiO_2$ and obtained an average interfacial shear stress of approximately 11.09 MPa between the $MoS_2$ monolayer and the $SiO_2$ substrate. This value is significantly larger than the shear stress between $MoS_2$ bilayers, indicating that the previous clamped-boundary model is rational for the $SiO_2$ substrate.

Moreover, we performed molecular dynamics (MD) simulations[42] to estimate the shear stress for $MoS_2@MoS_2$ and $MoS_2@SiO_2$. In our simulations, the upper $MoS_2$ monolayer is pulled along a certain direction on either a fixed $MoS_2$ monolayer or amorphous $SiO_2$ substrate. We took the average of the friction stress over time and obtained the average shear stress for $MoS_2@MoS_2$ and $MoS_2@SiO_2$. Details about MD simulations are supplied in the Supplementary Information. Notably, the average

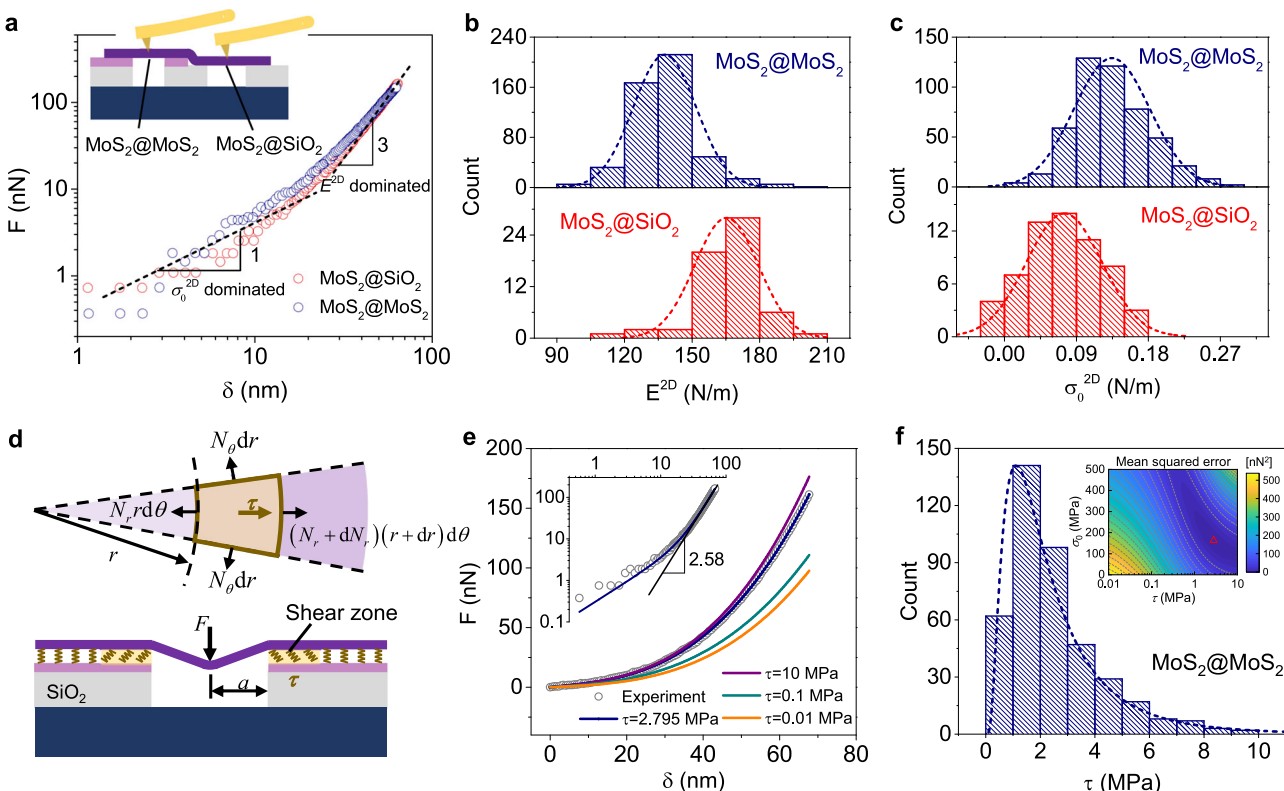

**Fig. 3 Determination of shear stress at TBLM interfaces with the shearing-boundary model. a** Typical force-indentation depth ($F$-$\delta$) curves of MoS$_2$@MoS$_2$ (blue dots) and MoS$_2$@SiO$_2$ (red dots) in logarithmic coordinates. The inset shows a schematic diagram of MoS$_2$@MoS$_2$ and MoS$_2$@SiO$_2$. **b**, **c** Histograms of the in-plane elastic modulus ($E^{2D}$) and pretension ($\sigma_0^{2D}$) of all samples measured over 1-μm-diameter and 1.5-μm-diameter holes. Here, MoS$_2$@MoS$_2$ samples exhibit smaller $E^{2D}$ and larger $\sigma_0^{2D}$ than MoS$_2$@SiO$_2$ samples. The dashed lines represent the fitting curves based on Gaussian distribution. **d** Schematic illustrations of the nanoindentation process on a 2D membrane and the proposed theoretical model that considers an interfacial shear zone between the tested membrane and the substrate. The in-plane equilibrium analysis for a representative element is illustrated in the top part. $N_r$ and $N_\theta$ are the radial and circumferential stress resultants, respectively. $r$ is the distance between the selected element and the center of the hole. $d\theta$ is the angle of the sector. $a$ is the radius of the hole. **e** Comparison between one typical experimental curve and fitted curves based on Eq. (2). The inset is the fitting curve based on $\tau = 2.795$ MPa, suggesting that when the interlayer shear stress $\tau = 2.795$ MPa, the fitting error reaches a minimum. **f** Distribution of fitted shear stresses that satisfies a logarithmic normal distribution. The inset shows a contour map of the nonlinear fitting error to obtain the closest shear stress of this sample.

shear stresses (2.51 MPa and 11.09 MPa) of MoS$_2$@MoS$_2$ and MoS$_2$@SiO$_2$ from our theoretical fitting are comparable to those (4.08 MPa and 13.69 MPa) from MD simulations, respectively. We also performed density functional theory (DFT) calculations to further characterize the interlayer shear stress for MoS$_2$@MoS$_2$. More details are given in the Supplementary Information. The average interlayer shear stress (4.87 MPa) along the minimum energy path from our DFT calculations is close to that (4.08 MPa) from our MD simulations. These results imply that our theoretical model can be used to estimate the interfacial shear stress between a thin membrane (even for 2D materials) and a substrate. However, there exists a certain error induced by using Eq. (2) to fit the experimental results, since Eq. (2) is an approximate solution for indentation of ultrathin elastic membrane with the shearing-boundary condition. The error might mainly originate from the approximation and simplification during the derivation of Eq. (2): (i) ignoring the finite size of indenter, (ii) simplifying nonlinear distribution of interlayer shear stress between the membrane and substrate, and (iii) simplifying complex coupling/interplay among in-plane stiffness, out-of-plane deflection, pretension, and interlayer shearing.

**Twist-angle independence of interlayer shearing.** Having established a reliable mechanical model, we calculated the

variation in shear stress by fitting the experimental data with different twist angles, as illustrated in Fig. 4a. The shear stress between the TBLM is independent of the twist angle, which is consistent with our previous hypothesis. This result can be explained by the fact that the overall interlayer vdW force is the sum of intermolecular forces, which homogenously spread over the interfaces of 2D materials. To complement the experimental results, we performed large-scale MD simulations for the nanoindentation of MoS$_2$ monolayers on MoS$_2$ and SiO$_2$ substrates. This time, we simulated the nanoindentation process instead of the planar friction to ensure that the setup of MD simulations was very similar to the experimental configuration, as shown in Fig. 4b. We also simulated the MoS$_2$ monolayer with different twist angles with respect to the MoS$_2$ substrate, as illustrated in Fig. 4c. More details of the MD simulations can be found in the Supplementary Information. Figure 4d shows some typical nanoindentation curves from MD simulations. In the large displacement regime, the scaling exponents of the nanoindentation force with respect to displacement are approximately 2.50 for MoS$_2$@MoS$_2$ and 2.74 for MoS$_2$@SiO$_2$, which are close to those of the experimental curves shown in Fig. 3a and Supplementary Fig. 5b. A similar phenomenon has been captured by our theoretical model, as evidenced in Fig. 3e. Such nonlinear behavior is attributed to the common contributions of the second and third terms on

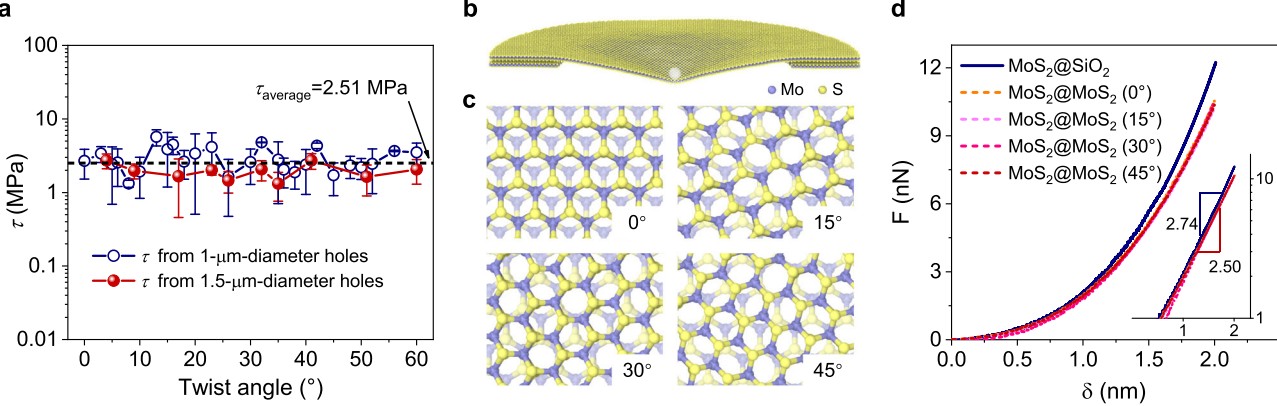

**Fig. 4 Twist-angle-independent shearing in TBLM. a** Independence of shear stresses of TBLM samples on twist angles. Error bars represent standard deviations. The dashed line marks the average $\tau$ ($\tau_{average}$) in all TBLM samples measured over 1-μm-diameter and 1.5-μm-diameter holes. **b** Atomic configurations of the simulated system. The white sphere represents a nanoindenter with a radius of 1 nm. **c** Top views of indented $MoS_2$ monolayer with different twist angles relative to the $MoS_2$ substrate. **d** Nanoindentation force–displacement ($F$-$\delta$) curves obtained from molecular dynamics simulations, also indicating that the interlayer mechanical interaction remains constant regardless of twist angles. The inset shows the scaling exponents of the force curves.

the right side of Eq. (2) from our theoretical model, where the second term is the membrane stretching related to the elastic modulus and the third term is related to the interfacial shear between the membrane and the substrate. Moreover, for different twist angles, the nanoindentation curves nearly coincide with each other. This result is consistent with the experimental results shown in Supplementary Fig. 9. The reason is that both moduli related to membrane stretching (i.e., the second term in Eq. (2)) due to nanoindentation and interfacial shear (i.e., the third term in Eq. (2)) are independent of the twist angle. The above results from our theoretical model and MD simulations suggest that the interfacial shear stress between the $MoS_2$ monolayer and the substrate significantly affects the nonlinear force–displacement behaviors, especially in the large deformation regime. Overall, our theoretical model considering interfacial shear stress can be used to accurately characterize the force–displacement relationship of $MoS_2$ monolayers on different substrates and to ensure the consistency of moduli measured from different substrates.

To conclude, we put forward an experimental configuration and a mechanical model to quantitatively determine the interlayer mechanical interaction of 2D materials. We discovered that the measured moduli and interlayer shear stress of $MoS_2$ are both independent of the twist angle. This can be attributed to the long-range interaction of vdW forces that homogenously spread over the interfaces of 2D materials. The shear stresses of the $MoS_2$–$MoS_2$ interface and $MoS_2$–$SiO_2$ interface are 2.51 and 11.09 MPa according to our experiments, which coincide well with the values obtained from MD simulations (4.08 and 13.69 MPa, respectively). Our strategy can be facilely applied to probe the interlayer mechanical interactions of other 2D systems and sheds light on experimentally obtaining the interfacial shear stress between 2D material interfaces.

## Methods

**Synthesis of $MoS_2$ monolayers.** $MoS_2$ monolayers were grown on $SiO_2$/Si substrates under ambient pressure in a chemical vapor deposition (CVD) system. A piece of $SiO_2$/Si substrate treated in piranha solution ($H_2SO_4$:$H_2O_2$ = 3:1) was faced down and placed on a quartz boat filled with $MoO_3$ powder. A droplet of PTAS solution was used as a seeding promoter. PTAS was synthesized by the alkaline hydrolysis of perylene-3,4,9,10-tetracarboxylic dianhydride (PTCDA). At first, KOH aqueous solution was added into the mixture of PTCDA and ethanol. Then, the reaction mixture was refluxed for 20 h. At last, the final

product PTAS was filtrated out after ethyl ether was added to the solution[43]. Before heating, the whole CVD system was purged with 200 sccm Ar for 10 min. Then, the temperature of $MoO_3$ was ramped to 650 °C at a rate of 15 °C/min and maintained at 650 °C for 3 min with 5 sccm Ar. The temperature of sulfur powder heated by a heating belt was ramped to 180 °C at a rate of 30 °C/min as soon as the temperature of $MoO_3$ reached 500 °C. After growth, the furnace was opened for rapid cooling.

**Preparation of TBLM.** For the wet transfer, a layer of PMMA was spin-coated with a speed of 2500 rpm on a $SiO_2$/Si substrate with triangular $MoS_2$ monolayers, which were used as bottom $MoS_2$ monolayers (Fig. 1a). Then, the PMMA/$MoS_2$ layer was etched away from the substrate in 1 M KOH solution. After rinsed with ultrapure water three times, the PMMA/$MoS_2$ layer was picked up by a clean holey substrate that was pre-etched by UV photolithography and dry-etched into patterns of holes with a diameter of ~1 μm or ~1.5 μm and a periodic interval of 2.5 μm. The holey substrate with the PMMA/$MoS_2$ layer was then heated at 180 °C for 1 min and immersed in acetone at 80 °C for 2 h to remove PMMA. After that, the $MoS_2$ monolayers over the holes collapsed. Finally, the holey substrate with the $MoS_2$ monolayers was annealed at 350 °C in vacuum ($1 \times 10^{-3}$ Pa) to further remove any PMMA residues.

For the dry transfer, an atomically flat Si wafer was used as a supporting substrate to cure PDMS, which avoids the large surface roughness of PDMS that may induce wrinkles on $MoS_2$ monolayers. The cured PDMS was then cut into a small piece and adhered onto a glass slide, and the glass slide/PDMS was further attached to a $SiO_2$/Si substrate with triangular $MoS_2$ monolayers, which were used as upper $MoS_2$ monolayers (Fig. 1a). The glass slide/PDMS/$MoS_2$ was removed from the $SiO_2$/Si substrate by immersing them in ultrapure water for one hour. Then, the glass slide/PDMS/$MoS_2$ was aligned with and adhered onto the bottom $MoS_2$ monolayers on the holey substrate by homemade transfer equipment in an Ar glove box. After heating at 60 °C for 15 min, the glass slide/PDMS was lifted upwards, leaving the upper $MoS_2$ monolayers stacked on top of the bottom $MoS_2$ monolayers and forming TBLM with random twist angles. Finally, the TBLM sample was annealed at 350 °C in vacuum ($1 \times 10^{-3}$ Pa) to remove PDMS residues on the surface and ensure that the twisted bilayer interacted effectively.

**Characterizations.** An optical microscope (OLYMPUS BX51 M) was used to find $MoS_2$ flakes and measure the twist angles of TBLM. Atomic force microscopy (AFM, Bruker Multimode 8) was used to measure surface topology and conduct nanoindentation tests. Raman spectra were obtained by a spectrometer (Horiba iHR550) using an excitation laser with a wavelength of 532 nm. ADF-STEM images of Moiré patterns were obtained on a JEM-ARM200CF operated at an acceleration voltage of 200 kV, with a collection angle of 40–160 mrad.

**Nanoindentation.** Before indentation, the sample was scanned in tapping mode under AFM until the thermal drift was negligible. Then, the AFM tip was positioned at the center of a suspended membrane. With the sample stage moving upwards by a distance of $z$, a point force $F_0$ was applied to the sample. Upon reaching the preinstalled force, the sample stage moved downwards to release the

force. After each series of nanoindentation tests, we scanned the sample again to detect any possible slippery.

**MD simulations**. MD simulations were performed via the large-scale atomic/molecular massively parallel simulator (LAMMPS)[42]. The details of the MD simulations are provided in the Supplementary Information.

**DFT calculations**. DFT calculations were performed via VASP[44] to characterize the interlayer shear stress for bilayer $MoS_2$. The details of the DFT calculations are provided in the Supplementary Information.

## Data availability
Relevant data supporting the key findings of this study are available within the article and the Supplementary Information file. All raw data generated during the current study are available from the corresponding authors upon request.

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

## Acknowledgements
We thank Prof. Liying Jiao for her help in sample growth, and Prof. Yang Wei and Prof. Zheng Han for their help in preparation of twisted samples. This work was financially supported by Basic Science Center Project of NSFC under grant No. 51788104, National Key R&D Program of China (2018YFA0208401), and National Natural Science Foundation of China under grant Nos. 51972193 and 91963117.

## Author contributions
K.L., Y.S., and X.L. conceived the project and designed the experiments. B.W. and Y.H.W. sythesized $MoS_2$ flakes. Y.S. and B.W. prepared TBLM samples and measured Raman spectra. Y.S. and E.W. performed nanoindentation experiments. Y.J.W. and X.L. performed the theoretical model and derived formula. H.Z. performed atomistic simulations. Y.Z. verified and analyzed simulation data. Q.Z. and L.G. carried out the ADF-STEM characterizations. Y.S., Y.J.W., K.L., and X.L. analyzed results and wrote the manuscript.

## Competing interests
The authors declare no competing interests.
