## [Peer Review File · Nature Communications]

Determining the Interlayer Shearing in Twisted Bilayer MoS₂ by NanoindentationREVIEWER COMMENTS

Reviewer #1 (Remarks to the Author):

The manuscript (MS) by Yufei Sun¹ et al. reports on the shear between molybdenum disulfide layers (MDS). The MS is an excellent experimental work well supported by analytical and computational contributions. The double-step procedure for sample preparation is simply brilliant. The data statistic is also superb, so my zero-order response is yes, the MS is suitable for publication in Nature Communications.

However, I have a “but” related with figure 1d that shows an AFM topography of a drumhead covered by a single layer of MDS. As usual, MDS flakes present a random distribution of bubbles and wrinkles covering a non-negligible surface area of the image. The implication is that as the surface area of MDS in actual contact is much lower than the one derived for perfectly flat layers without bubbles, the figure obtained for the interfacial shear stress modulus can be just a lower bound for the actual magnitude. The origin of these bubbles and wrinkles is not clear, but I suspect they are related to the surface conditions and the transfers method. In fact, the adhesion between layers upon transfer can be improved by exerting high pressure by a diamond AFM tip (see Yolanda Manzares-Negro et al.)https://scholar.google.es/citations?user=TbzF2VM AAAAJ&hl=es#d=gs_md_cita-d&u=/citations%3Fview_op=view_citation%26hl=es%26user=TbzF2VM AAAAJ%26citation_for_view=TbzF2VM AAAAJ%3Au-x6o8ySG0sC%26tzm=-120.

The experiments reported there support my assumption that the MS values for the shear modulus can be just a lower bound for the actual value of this elastic constant. I want the authors to consider this issue carefully and pay attention to its implications before publication.

I have other minor points that I listed below:

- DFT simulations might help to validate the value reported for the interfacial shear stress.
- I think is a good idea to follow the same scheme as in the original paper by Hone and coworkers. Move fig 2b to the SI and leave just a Force vs delta in the main. Force vs. distance curves are important but rather technical in this context.
- Did you measure indentation near the breaking force?. Please comment.
- Did you observe variations in E or tau with time (days, weeks, or even months).

Reviewer #2 (Remarks to the Author):

In this work, Sun et al. detected 2D interfacial mechanical interactions of twisted bilayer MoS₂ with nano-indentation method and demonstrated their independence of the twist angle. If the commonly used fixed-boundary mechanical model was considered, the measured in-plane elastic modulus is ~23% smaller than the intrinsic value. In view of this, the authors established a new shearing-boundary model by considering a finite interfacial shear stress between the tested membrane and the substrate. In the end, the twist angle independence of in-plane elastic moduli and interlayer shear stress was explained by molecular dynamic stimulations.

The study is interesting and well-organized and the results can be published. But, some concerns need to be clarified before publication.

1. The authors claimed the observation of twist-angle-independence of interlayer shearing which is attributed to the long-range interaction of intermolecular vdW forces that homogeneously spread over the interfaces of 2D materials. Is this conclusion credible considering the large data dispersion (Fig. 2c and Fig. 4a)? Besides, the twist-angle-dependence of interlayer friction of 2D materials has been measured, at least for graphene (doi.org/10.1021/acsnano.7b09083). The authors should explain why and make detailed comparison to previous results.
2. Twist angle describes the lattice mismatch between the double layers. In this work, nano-indentation may induce in-plane deformation in the top layer, or mismatch of lattice to the bottom layer. This deformation induced lattice mismatch should be more complicated than the twist angle

induced lattice mismatch. In this consideration, is “Twist-Angle-Independent Interlayer Shearing in Twisted Bilayer MoS₂” an accurate description of the observed phenomenon?

3. The nano-indentation method is an indirect method of measuring the mechanical properties which relies heavily on the mechanical model. In this work, the authors established a new shearing-boundary model. But the new model needs to be validated besides the obtained smaller shear stress of MoS₂@MoS₂ compared to MoS₂@SiO₂ (manuscript, lines 241-246). At least, the measured in-plane elastic modulus of MoS₂ based on the shearing-boundary model should be provided to clarify its advance compared to previous fixed-boundary model.

Minor technique issues:

4. The reproducibility of measured data should be clearly exhibited. For example, the number of samples and repeated measurements of F- δ curves.

5. The interfacial cleanliness of bilayer MoS₂ and the edge of the broken bottom MoS₂ flake which is crucial for clarifying the large data dispersion should be further characterized.

6. The language needs improvement. For example, Line 30: “independent on” should be “independent of”; Line 33: “suggesting” should be “suggesting that”...

Reviewer #3 (Remarks to the Author):

This work by Yufei Sun et al. reports the measurement of interlayer shearing stress in twisted bilayer MoS₂ using an indentation-based method. Shearing at 2D material interfaces is of importance for sure as it influences the system in various ways. However, I found a number of significant flaws in experiments, analytical theory, and modelling, which would make the finding of this work very questionable. I am afraid that I need to reject the publication in Nature Communications. At the same time, I would provide the following comments/suggestions that I hope to be helpful.

1. When talking about twisted bilayers, it is really important to assure samples are clean. However, what I found particularly in Supplementary Figure 2 is that the interlayer interfaces have been contaminated seriously – full of wrinkles, bubbles, and so on. With such contamination, it is not fair to claim what has been measured is truly related to the twist-angle-related interface! Better to prepare samples with clear moiré signals.

2. Using indentation to extract stiffness of 2D elastic sheets is not a reliable way though this erroneous method has been propagating in the past decade. First of all, the pretension term in eq (1) is wrong – the asymptotic behaviour suggests a logarithmic dependence on the tip radius so that smaller radii turn out to make a larger contribution (see Vella & Davidovitch 2017). Second, the error produced by the sum of pretension and membrane tension could be on the order of 1 – this ad hoc sum can not be used at all.

3. The authors discussed two hypotheses for their finding on the angle-independent moduli. Again, I can not be convinced by the finding itself given the contaminated samples and erroneous fitting method used. Besides, both hypotheses have been explained well in some recent works, in particular Davidovitch & Guinea 2021 and Dai and Lu 2021. What appeared more important is the presence of wrinkling due to the sliding of the vdW interface, as observed in Pablo Ares et al PNAS 2021. The wrinkling would further modify the mechanical response to a point load in a significant manner but is not considered here. Indeed, it is a mess/impossible to consider those important things in one simple expression, which makes the indentation methodology less useful.

4. Eq (2) proposed by the authors is wrong as well in several aspects. First, the pretension term is incorrect as mentioned above; Second, the superposition of three terms for a nonlinear problem is wrong in itself. Three, the deriving of the third used a wrong boundary condition (supplementary eq 8 and 9 where the infinity is subject to a pretension instead of null). Four, thin sheets like MoS₂ cannot sustain any compressive force so the formation of instabilities makes the analysis inside and outside the hole inappropriate.

5. In MD simulations, LJ potential used here is not a good choice (even erroneous) to elucidate the angle-dependent interactions. KC is way better. To better match what is going on in experiments, an axisymmetric model is recommended, which may be used to verify equation (2) since it is the key to the measurement.

Reviewer #1 (Remarks to the Author):

[General Comment] The manuscript (MS) by Yufei Sun et al. reports on the shear between molybdenum disulfide layers (MDS). *The MS is an excellent experimental work well supported by analytical and computational contributions. The double-step procedure for sample preparation is simply brilliant. The data statistic is also superb, so my zero-order response is yes, the MS is suitable for publication in Nature Communications.*

[Response] We greatly appreciate the reviewer for her/his taking time to review our paper and her/his evaluation of our work. In our revised manuscript, we have performed more reliable experiments and calculations to support our statements. The following is a point-by-point response to the specific comments.

[Comment] However, I have a “but” related with figure 1d that shows an AFM topography of a drumhead covered by a single layer of MDS. As usual, MDS flakes present a random distribution of bubbles and wrinkles covering a non-negligible surface area of the image. The implication is that as the surface area of MDS in actual contact is much lower than the one derived for perfectly flat layers without bubbles, the figure obtained for the interfacial shear stress modulus can be just a lower bound for the actual magnitude. The origin of these bubbles and wrinkles is not clear, but I suspect they are related to the surface conditions and the transfers method. In fact, the adhesion between layers upon transfer can be improved by exerting high pressure by a diamond AFM tip (see Yolanda Manzares-Negro et al). The experiments reported there support my assumption that the MS values for the shear modulus can be just a lower bound for the actual value of this elastic constant. I want the authors to consider this issue carefully and pay attention to its implications before publication.

[Response] We thank the reviewer for the valuable comment. We agree that the bubbles and wrinkles existing at the twisted bilayer interfaces are related to surface conditions and transfer methods. Large quantities of bubbles and wrinkles would reduce the contact area between twisted bilayers and lead to unpredictable influence on the interlayer interactions. Therefore, cleaner interfaces are more preferred.

We also thank the reviewer for providing an approach to improving the interlayer adhesion according to the reference (Yolanda Manzares-Negro et al, *ACS Appl. Mater. Interfaces* 2020, 12, 37750). In this method, a diamond AFM tip was used to exert a high pressure on a monolayer graphene flake, improving the graphene-substrate adhesion in a square rim area to seal the gas inside a drumhead. However, it is regret that this method is not applicable to our experiment, because it cannot be used to precisely seal the circular rim around the edge of each drumhead and it will take an unacceptable long time to seal tens of drumheads that are necessary for our statistical analysis.

In the past several months, we have made many efforts to minimize the density of bubbles and wrinkles by keeping our transfer processes clean in all aspects. The details are described in the Methods section of the revised manuscript. In our optimized transfer method, we used high-quality and clean MoS₂ triangular single-crystal monolayers (Fig. R1a). Our transfer processes still include a PMMA-assistant wet transfer to break the bottom MoS₂ monolayer region over holes and a PDMS-assistant dry transfer to stack and suspend the upper MoS₂ monolayer region over holes, as shown in Fig. 1a. This random stacking is very efficient for preparing twisted MoS₂ bilayers with various twist angles. We also tried to reduce the data dispersion by further using holey substrates with larger

holes ($\sim 1.5 \mu\text{m}$ in diameter) in the nanoindentation to reduce the relative offset of indentation positions and the deviation of measured hole sizes.

Fig. R1 | As-grown (a) and transferred (b, c) MoS₂ triangular monolayers. Clean surfaces and few bubbles/wrinkles can be observed on MoS₂ monolayers after either optimized PMMA-assistant wet transfer (b) or optimized PDMS-assistant dry transfer (c).

Specifically, in the wet transfer, we kept each solution process clean and annealed the bottom MoS₂ monolayers at 350 °C in vacuum ($\sim 1 \times 10^{-3}$ Pa) to remove any PMMA residues. As shown in Fig. R1b, a MoS₂ monolayer transferred onto a flat SiO₂/Si substrate after this step show a very clean surface that is similar to that of the SiO₂/Si substrate. In the dry transfer, PDMS was cured on an atomically flat Si wafer, which avoids the large surface roughness of PDMS that may induce wrinkles of MoS₂ monolayer. Then the dry transfer of the upper MoS₂ monolayer was done in an Ar glove box to avoid interface contaminations from air during the twisted stacking process. At last, the sample was annealed again to remove PDMS residues on the surface and have the twisted bilayer interact effectively. Figure R1c shows a MoS₂ monolayer dry transferred onto a holey SiO₂/Si substrate, which also has a clean surface with few bubbles and wrinkles.

With these efforts, we have effectively reduced the density of bubbles and wrinkles in the twisted region and obtained clean samples on holey substrates (Fig. R2a-c). Compared with the large quantity of bubbles and wrinkles shown in our original manuscript (Fig. R2d and e), the bubbles and wrinkles have been greatly eliminated in the twisted region after our optimized clean transfer processes (Fig. R2b and c).

Despite the achievement of clean samples, we note that a thorough elimination of bubbles and wrinkles in the twisted region of 2D materials seems impossible, because in this region the 100% clean interface cannot be obtained. Figure R3 shows a comparison of bubbles coverage ratio between our twisted samples and other reported results. The overall coverage ratio of bubbles and wrinkles in our sample has been close to the lows in the twisted samples reported in literature (e.g., for thick samples, *Nat. Nanotech.* 2021, **16**, 888, *Nat. Commun.* 2019, **10**, 2302; for monolayer samples, *Nat. Commun.* 2020, **11**, 2153). Considering that the materials we twisted are monolayers and the target substrate is holey, both of which are much easier to cause bubbles or wrinkles than thick flakes or flat substrates, we believe that our twisted samples have already been of very high quality. The clear Moiré patterns observed under annular dark-field scanning transmission electron microscopy (ADF-STEM) also suggest the high quality and clean interfaces of our twisted samples (Fig. R4).

Fig. R2 | Twisted MoS₂ bilayers on holey substrates. (a) Optical image of a PDMS-transferred upper MoS₂ monolayer (yellow dashed area) overlapping a PMMA-transferred bottom MoS₂ monolayer (white dashed area), forming a twisted region. (b, c) AFM images of a twisted MoS₂ region on a holey substrate made by the optimized transfer processes. (d, e) AFM images of a twisted MoS₂ region made by the previous transfer processes in our original manuscript.

Fig. R3 | Comparison of our twisted samples on holey substrates with others on flat substrates reported in literature. A very low density of bubbles and wrinkles is observed in the twisted region of our monolayer samples on a holey substrate. The bubbles coverage ratio of our samples has reached the lows as reported on flat substrates.

Fig. R4 | Clear Moiré patterns observed under ADF-STEM in our twisted bilayer MoS₂ samples with twist angles of 24° (a) and 9° (b). Insets show the corresponding simulated Moiré patterns, well consistent with the experimental observations.

We also note that in our nanoindentation experiment, the suspended upper MoS₂ monolayer is clamped somewhere around the hole edge according to the fixed-boundary model or shearing-boundary model. Therefore, the measured moduli would strongly depend on the tightness of the twisted region around the hole edge. This region relatively lacks bubbles and wrinkles because they may merge and move inside the hole. As shown in the most left panel of Fig. R3, the coverage ratio of bubbles and wrinkles in the 200-nm-width rim of a hole edge is only ~10 %, reaching the lowest level reported on flat substrates. With the improved sample quality, our new samples exhibit a modulus of ~165 N/m for MoS₂@SiO₂ and moduli ranging from 132 to 144 N/m for MoS₂@MoS₂ (Fig. R5, red data). The difference in the moduli of MoS₂@SiO₂ and MoS₂@MoS₂ is ~20 %, apparently larger than the error possibly induced by the bubbles coverage (~10 %), suggesting that our new results are more solid for the statement of the shearing-boundary model. Note that the new measured moduli of MoS₂@SiO₂ and MoS₂@MoS₂ are very close to our previous results although the new data have much smaller deviation (Fig. R5), which indicates the reliability of our previous results and validates our statement that only a small twisted area around the hole edge affects the moduli measurements.

Fig. R5 | Measured moduli. The MoS₂@MoS₂ moduli obtained from 1-μm-diameter holes are

previous results (blue circles and blue stars), while the moduli from 1.5- μm -diameter holes are new results for the samples prepared by the optimized transfer processes (red circles and orange stars).

In short summary, we have greatly improved our transfer processes to reduce the contamination and minimize the density of bubbles and wrinkles. The new data are consistent with our previous results and support the shearing-boundary model in twisted MoS_2 bilayers.

In response to this comment, we added Fig. R1 as Supplementary Fig. 1, Fig. R2a and R2c as Fig. 1b and 1c, Fig. R3 as Supplementary Fig. 3, Fig. R4 as Fig. 1d, and Fig. R5 as Fig. 2c. We also cited the suggested reference, supplemented the above discussion on page 6, 7, and 10-12 of the revised manuscript, and described the preparation of the twisted samples in detail on page 19 and 20 in Methods section.

[Comment] I have other minor points that I listed below:

DFT simulations might help to validate the value reported for the interfacial shear stress.

[Response] We thank the reviewer for her/his helpful suggestions. Following the suggestion of the reviewer, we performed density function theory (DFT) calculations via VASP to characterize the interlayer shear for bilayer MoS_2 . According to the method reported in the previous reference (*Phys. Rev. B*, 92, 085434, 2015), we calculated the work of separation ΔW_{sep} and friction force f for the sliding of the upper layer with respect to the lower layer. Figure R6a and b show the variation of work of separation with the sliding distance along the armchair direction and the minimum energy path (MEP), respectively. The results from our DFT calculations agree well with those from previous DFT calculations (*Phys. Rev. B*, 92, 085434, 2015) via QUANTUM ESPRESSO. Figure R6c and d show the variation of friction force with the sliding distance along the armchair direction and MEP, respectively. We took average of the friction force over the sliding distance, then divided by the unit cell area, and obtained the average interlayer shear stress for bilayer MoS_2 for armchair direction and MEP as 8.81 MPa and 4.87 MPa, respectively. These values of average interlayer shear stress are close to that (4.08 MPa) from our MD simulations.

In response to this comment, we have added the following sentences on page 15 and 16 to introduce our DFT calculations for interlayer shear stress for bilayer MoS_2 ,

“We also performed density functional theory (DFT) calculations to further characterize the interlayer shear stress for $\text{MoS}_2@ \text{MoS}_2$. More details are given in the Supplementary Information. The average interlayer shear stress (4.87 MPa) along the minimum energy path from our DFT calculations is close to that (4.08 MPa) from our MD simulations.”

We also added a brief introduction of our DFT calculations in the Method section (page 21), and more details of our DFT calculations and associated results as the part 6 in Supplementary Information.

Fig. R6 | Work of separation and friction force during sliding. (a,b) Variation of work of separation with the sliding distance along the armchair direction and MEP. The red lines are from our DFT calculations, while the black lines are from previous DFT calculations (*Phys. Rev. B*, 92, 085434, 2015). (c,d) Variation of friction force with the sliding distance along the armchair direction and MEP.

[Comment] I think is a good idea to follow the same scheme as in the original paper by Hone and coworkers. Move fig 2b to the SI and leave just a Force vs delta in the main. Force vs. distance curves are important but rather technical in this context.

[Response] We thank the reviewer for the suggestion. In the revised manuscript, we only left the F - δ curve in Fig. 2b, and show the F - z curve in Supplementary Fig. 5a.

[Comment] Did you measure indentation near the breaking force? Please comment.

[Response] We thank the reviewer for the comment. In our original manuscript, we performed the indentation under moderate loads (90-450 nN) to avoid the unrecoverable deformation of samples (e. g. plastic deformation or fracture). Following the reviewer's suggestion, we further measured the indentation near the breaking force. Fig. R7 shows that the F - δ curve of $\text{MoS}_2@/\text{SiO}_2$ or $\text{MoS}_2@/\text{MoS}_2$ follows the nearly identical trace under different loads and still obeys the cubic relationship ($F \sim \delta^3$) near the breaking force, indicating a brittle fracture behavior and very good reproducibility of the force curves.

Fig. R7 | F - δ curves of MoS₂@MoS₂ (a) and MoS₂@SiO₂ (b) under different loads.

In response to this comment, we have added Fig. R7 as Supplementary Fig. 6 and the above description in the figure caption as well as on page 10 of the revised manuscript.

[Comment] Did you observe variations in E or τ with time (days, weeks, or even months).

[Response] We are grateful to the reviewer for the comment. We performed the nanoindentation and measured E^{2D} and τ every a few days in two months. It was found that both E^{2D} and τ do not apparently change with the time (see Fig. R8, E^{2D} varies by <5% and τ from 1.3 to 4.3 MPa) in two months. We have added Fig. R8 as Supplementary Fig. 7 and the related description on page 10 of the revised manuscript.

Fig. R8 | Variations of E^{2D} and τ with time in two months.

Reviewer #2 (Remarks to the Author):

[General Comment] In this work, Sun et al. detected 2D interfacial mechanical interactions of twisted bilayer MoS₂ with nano-indentation method and demonstrated their independence of the twist angle. If the commonly used fixed-boundary mechanical model was considered, the measured in-plane elastic modulus is ~23% smaller than the intrinsic value. In view of this, the authors established a new shearing-boundary model by considering a finite interfacial shear stress between the tested membrane and the substrate. In the end, the twist angle independence of in-plane elastic moduli and interlayer shear stress was explained by molecular dynamic stimulations. *The study is interesting and well-organized and the results can be published.* But, some concerns need to be clarified before publication.

[Response] We greatly appreciate the reviewer for her/his taking time to review our paper and her/his evaluation of our work. In the revised manuscript, we have performed more reliable experiments and calculations to support our statements. The following is a point-by-point response to the reviewer's specific comments.

[Comment] 1. The authors claimed the observation of twist-angle-independence of interlayer shearing which is attributed to the long-range interaction of intermolecular vdW forces that homogeneously spread over the interfaces of 2D materials. Is this conclusion credible considering the large data dispersion (Fig. 2c and Fig. 4a)? Besides, the twist-angle-dependence of interlayer friction of 2D materials has been measured, at least for graphene (doi.org/10.1021/acsnano.7b09083). The authors should explain why and make detailed comparison to previous results.

[Response] We thank the reviewer for the valuable comment. Yes, we have considered the data dispersion for the conclusion. In mechanical measurements of nanoscale materials, large data dispersion is usually unavoidable. The data dispersion originates from many aspects in our experiments, such as the difference between single-crystal flakes, the offset of indentation positions, and the deviation of measured hole sizes. Therefore statistical data are indispensable for a solid conclusion. In our previous experiments, we measured more than 100 samples with 26 twist angles, and all conclusions are based on the statistical results.

In the past several months, we have tried two ways to further reduce the data dispersion. First, we optimized our transfer processes in all aspects (see details on page 18 and 19 in the Methods section) to avoid bubbles or wrinkles in the twisted region of samples (Fig. R2). Compared with those shown in our original manuscript (Fig. R2d and e), the bubbles and wrinkles have been greatly eliminated in the twisted region after our optimized clean transfer processes (Fig. R2b and c). Second, we used high-quality single-crystal MoS₂ monolayers and larger holes (~1.5 μm in diameter) in the nanoindentation to reduce the relative offset of indentation positions and the deviation of measured hole sizes. As shown in Fig. R4, our new samples exhibit a modulus of ~165 N/m for MoS₂@SiO₂ and moduli ranging from 132 to 144 N/m for MoS₂@MoS₂. The deviation of the statistical moduli for each twist angle ranges from 2% to 8%, much smaller than the difference of the moduli between MoS₂@SiO₂ and MoS₂@MoS₂ (~20 %), suggesting that our new data are quite reliable. Note that the new measured moduli of MoS₂@SiO₂ and MoS₂@MoS₂ are very close to our previous results (Fig. R5), which indicates the reliability of our previous results that were measured over 1-μm-diameter holes.

Fig. R2 | Twisted MoS₂ bilayers on holey substrates. (a) Optical image of a PDMS-transferred upper MoS₂ monolayer (yellow dashed area) overlapping a PMMA-transferred bottom MoS₂ monolayer (white dashed area), forming a twisted region. (b, c) AFM images of a twisted MoS₂ region on a holey substrate made by the optimized transfer processes. (d, e) AFM images of a twisted MoS₂ region made by the previous transfer processes in our original manuscript.

Fig. R5 | Measured moduli. The MoS₂@MoS₂ moduli obtained from 1-μm-diameter holes are previous results (blue circles and blue stars), while the moduli from 1.5-μm-diameter holes are new results for the samples prepared by the optimized transfer processes (red circles and orange stars).

We also carefully read the reference suggested by the reviewer and compared it with our results. In the reference, the authors coated graphite layers onto an AFM tip and scanned the tip on a graphite flake to obtain the frictional characteristics. There are several notable differences between our work and the suggested one. First, they measured the friction at the slipping interface while we measured the shearing interaction between monolayers. Second, their friction measurement was conducted by a point contact between the friction pairs, but our work studied the overall interface shearing of 2D materials. Third, their friction measurement was based on uniaxial slipping because the friction of

the tip can only follow a single direction, while our experiments applied biaxial shearing at the interfaces. Lastly, although the authors observed two narrow peaks of high friction at certain “rotation angle”, they cannot quantitatively obtain the twist-angle-dependence of interlayer friction because the rotation angle is “a relative value where the absolute value does not have a physical meaning”; our work, however, exhibited a clear relationship between the interlayer shear moduli and the twist angle. Therefore, the two studies relate to different physical properties and cannot be simply compared.

In response to this comment, we added Fig. R2a and R2c as Fig. 1b and 1c, Fig. R5 as Fig. 2c, and part of the above discussion on pages 6, 7, and 10-12 of the revised manuscript. We also cited the suggested reference in the revised manuscript.

[Comment] 2. Twist angle describes the lattice mismatch between the double layers. In this work, nano-indentation may induce in-plane deformation in the top layer, or mismatch of lattice to the bottom layer. This deformation induced lattice mismatch should be more complicated than the twist angle induced lattice mismatch. In this consideration, is “Twist-Angle-Independent Interlayer Shearing in Twisted Bilayer MoS₂” an accurate description of the observed phenomenon?

[Response] We thank the reviewer for this comment. We agree that the in-plane deformation may occur in the nanoindentation experiment and this deformation is rather complicated. However, in order to avoid the unrecoverable deformation of samples (e. g. plastic deformation or fracture), we used the nanoindentation data obtained at moderate loads (90-450 nN). In this case, the corresponding indentation depth is much smaller than the diameter of holes (30-70 nm for holes in 1 μm diameter and 60-130 nm for holes in 1.5 μm diameter), and as a result, the strain applied on the upper MoS₂ monolayer is estimated to be less than 2% for all of the nanoindentation measurements. Under such small strains, the in-plane deformation of the upper monolayer is very minor compared with the twist-angle-induced lattice mismatch. Therefore, we did not consider the in-plane deformation in our experiments and theoretical simulation. And in this regard, the description “Twist-Angle-Independent Interlayer Shearing in Twisted Bilayer MoS₂” should be generally accurate. However, in order to better exhibit our method rather than describe the observed phenomenon, we have changed our title as “Determining the Interlayer Shearing in Twisted Bilayer MoS₂ by Nanoindentation”, and added the above discussion on page 10 of the revised manuscript.

[Comment] 3. The nano-indentation method is an indirect method of measuring the mechanical properties which relies heavily on the mechanical model. In this work, the authors established a new shearing-boundary model. But the new model needs to be validated besides the obtained smaller shear stress of MoS₂@MoS₂ compared to MoS₂@SiO₂ (manuscript, lines 241-246). At least, the measured in-plane elastic modulus of MoS₂ based on the shearing-boundary model should be provided to clarify its advance compared to previous fixed-boundary model.

[Response] We thank the reviewer for her/his valuable comments and suggestions. In our current study, the in-plane modulus E^{2D} of MoS₂@SiO₂, which is obtained based on the fixed-boundary model, is considered as the intrinsic modulus of MoS₂ as it is consistent with the values in previous literature and theoretical predictions. The measured in-plane elastic modulus of MoS₂@MoS₂, however, is softened and thus considered not intrinsic. In our original manuscript, we fixed our measured intrinsic modulus of MoS₂@SiO₂ as the in-plane modulus of MoS₂, and then used the shearing-boundary model to extract other two parameters (interlayer shear stress and pretension.

Following the suggestions of the reviewer, we made attempt to fit all three parameters (including in-plane modulus, interlayer shear stress and pretension) by using the shearing-boundary model. During fitting, the values of all three parameters are given only when the minimal fitting error between Eq. (2) and experimental force-displacement curve reaches. The average values of three parameters (including in-plane modulus E , interlayer shear stress τ and pretension σ_0) fitting over all tested curves are 253 GPa, 5.6 MPa and 176.9 MPa, respectively. These values are close to those reported in our original manuscript. It is noted that the average modulus obtained from nanoindentation of MoS₂@MoS₂ is smaller than that (266 GPa) from nanoindentation of MoS₂@SiO₂. But considering that the modulus is a material constant, we thought that during fitting via the shearing-boundary model, it is better and more reasonable to fix the in-plane modulus of MoS₂.

In response to this comment, we have added the following sentences on page 14 to address the details about modulus during numerical fitting,

“When we use Eq. (2) to fit the experimental curves, the modulus of MoS₂ is fixed and taken as the measured average value from nanoindentation for MoS₂@SiO₂ since the modulus is a material constant. Here, we mainly extract the interlayer shear stress by using Eq. (2) to fit the experimental curve. Undoubtedly, one can extract all three parameters (including modulus, interlayer shear stress and pretension) via the nonlinear fitting method.”

In fact, the shearing-boundary model is more general compared with the fixed-boundary model. Notably, when the interlayer shear stress τ tends to infinity in Eq. (2), our shearing-boundary model can be degraded into the fixed-boundary model. It indicates that the fixed-boundary model is only a limit case of our shearing-boundary model. To further validate our shearing-boundary model, we used our shearing-boundary model to fit the nanoindentation force-displacement curves of graphene on SiO₂ reported in ref [37] (*Science*, 2008, 321, 385). The obtained values of τ , σ_0 , and E for 1.5- μ m-diameter hole nanoindentation are 3.2 MPa, 810.6 MPa, and 1127.3 GPa, respectively. The obtained values of τ , σ_0 , and E for 1- μ m-diameter hole nanoindentation are 6.8 MPa, 1061.4 MPa, and 1183.6 GPa, respectively. These values of σ_0 and E are close to those reported in ref [37] (*Science*, 2008, 321, 385). Especially, the value of shear stress τ for graphene on \square SiO₂ substrate from 1.5- μ m-diameter hole nanoindentation is close to that (1-3 MPa) measured by pressurized microscale bubbling (*Phys Rev Lett*, 2017, 119, 036101). These results indicate the validity of our shearing-boundary model and also clarify its advance compared to previous fixed-boundary model. In response to this comment, we have added the above discussions as the part 7 in the revised Supplementary Information to address the validity of our model.

Minor technique issues:

[Comment] 4. The reproducibility of measured data should be clearly exhibited. For example, the number of samples and repeated measurements of F- δ curves.

[Response] We measured 113 samples with 26 twist angles over 1- μ m-diameter holes in our previous measurements, and further measured 46 samples with 10 twist angles over larger, 1.5- μ m-diameter holes in the new measurements (see Fig. R5). On each sample, we performed 3-5 consecutive nanoindentations under different loads, which means that we actually performed 3-5 repeated measurements. Fig. R7 shows that the F- δ curves of MoS₂@SiO₂ or MoS₂@MoS₂ in 5 consecutive nanoindentations under different loads follow the nearly identical trace until the

breaking of the upper MoS₂ monolayer, suggesting very good reproducibility of our measurements.

Fig. R7 | F- δ curves under different maximum forces.

In response to this comment, we have added Fig. R7 as Supplementary Fig. 6 and the above description in the figure caption as well as on page 10 of the revised manuscript.

[Comment] 5. The interfacial cleanliness of bilayer MoS₂ and the edge of the broken bottom MoS₂ flake which is crucial for clarifying the large data dispersion should be further characterized.

[Response] We thank the reviewer for pointing out the cleanliness issue of our samples. In response to this comment, we have optimized the transfer processes in all aspects to make the interface as clean as possible, and characterized the edge of the broken bottom MoS₂ monolayer. The data dispersion has been much reduced as mentioned in the response to the Reviewer's comment 1.

In our optimized transfer processes, we used high-quality and clean MoS₂ triangular single-crystal monolayers (Fig. R1a). Our transfer processes still include a PMMA-assistant wet transfer to break the bottom MoS₂ monolayer region over holes and a PDMS-assistant dry transfer to stack and suspend the upper MoS₂ monolayer region over holes, as shown in Fig. 1a (see details in the Methods section). This random stacking is very efficient for preparing twisted MoS₂ bilayers with various twist angles.

Specifically, in the wet transfer, we kept each solution process clean and annealed the bottom MoS₂ monolayers at 350 °C in vacuum ($\sim 1 \times 10^{-3}$ Pa) to remove any PMMA residues. As shown in Fig. R1b, a MoS₂ monolayer transferred onto a flat SiO₂/Si substrate after this step show a very clean surface that is similar to that of the SiO₂/Si substrate. In the dry transfer, PDMS was cured on an atomically flat Si wafer, which avoids the large surface roughness of PDMS that may induce wrinkles of MoS₂ monolayer. Then the dry transfer of the upper MoS₂ monolayer was done in an Ar glove box to avoid interface contaminations from air during the twisted stacking process. At last, the sample was annealed again to remove PDMS residues on the surface and have the twisted bilayer interact effectively. Figure R1c shows a MoS₂ monolayer dry transferred onto a holey SiO₂/Si substrate, which also has a clean surface with few bubbles and wrinkles.

Fig. R1 | As-grown (a) and transferred (b, c) MoS₂ triangular monolayers. Clean surfaces and few bubbles/wrinkles can be observed on MoS₂ monolayers after our optimized PMMA-assistant wet transfer (b) and PDMS-assistant dry transfer (c).

With these efforts, we have effectively improved the cleanliness in the twisted bilayer samples on holey substrates. The overall coverage ratio of bubbles and wrinkles in our sample has been close to the lows in the twisted samples reported in literature (e.g., for thick samples, *Nat. Nanotech.* 2021, **16**, 888, *Nat. Comm.* 2019, **10**, 2302; for monolayer samples, *Nat. Comm.* 2020, **11**, 2153). Considering that the materials we twisted are monolayers and the target substrate is holey, both of which are much easier to cause bubbles or wrinkles than thick flakes or flat substrates, we believe that our twisted samples have already been of very high quality. The clear Moiré patterns observed under annular dark-field scanning transmission electron microscopy (ADF-STEM) also suggest the high quality and clean interfaces of our twisted samples (Fig. R4).

Fig. R3 | Comparison of our twisted samples on holey substrates with others on flat substrates reported in literature. A very low density of bubbles and wrinkles is observed in the twisted region of our monolayer samples on a holey substrate. The bubbles coverage ratio of our samples has reached the lows as reported on flat substrates.

Fig. R4 | Clear Moiré patterns observed under ADF-STEM in our twisted bilayer MoS₂ samples with twist angles of 24° (a) and 9° (b). Insets show the corresponding simulated Moiré patterns, well consistent with the experimental observations.

We also note that in our nanoindentation experiment, the suspended upper MoS₂ monolayer is clamped somewhere around a hole edge according to the fixed-boundary model or shearing-boundary model. Therefore, the measured moduli would strongly depend on the tightness of the twisted region around the hole edge. This region relatively lacks bubbles and wrinkles because they may merge and move inside the hole. As shown in the most left panel of Fig. R3, the coverage ratio of bubbles and wrinkles in the 200-nm-width rim of a hole edge is only ~10 %, reaching the lowest level reported on flat substrates (Fig. R3, right panel).

At last, following the reviewer's suggestion, we characterized the hole edge of the broken bottom MoS₂ monolayer. As shown in Fig. R9, after the wet transfer, the bottom MoS₂ monolayers collapse over holes and exhibit sharp edges around the holes, while only the supported region remains on the substrate. It suggests that the data dispersion in moduli does not result from the hole edge of the broken bottom MoS₂ monolayer.

Fig. R9 | AFM image of a broken bottom MoS₂ monolayer over a hole on a SiO₂/Si substrate.

In short summary, we have optimized the transfer processes in all aspects to avoid contamination and make the interface at the twisted region as clean as possible. We have also characterized the edge of the broken bottom MoS₂ monolayer.

In response to this comment, we added Fig. R1 as Supplementary Fig. 1, Fig. R3 as Supplementary Fig. 3, Fig. R4 as Fig. 1d, and Fig. R9 as Supplementary Fig. 2. We also added the above discussion on page 6 and 7 of the revised manuscript and described the preparation of the twisted samples in detail on page 19 and 20 in Methods section.

[Comment] 6. The language needs improvement. For example, Line 30: “independent on” should be “independent of”; Line 33: “suggesting” should be “suggesting that”...

[Response] We thank the reviewer for the comment. These mistakes and errors have been corrected. We have also asked the Spring Nature Author Services to improve the language of the manuscript.

Reviewer #3 (Remarks to the Author):

[General Comment] This work by Yufei Sun et al. reports the measurement of interlayer shearing stress in twisted bilayer MoS₂ using an indentation-based method. Shearing at 2D material interfaces is of importance for sure as it influences the system in various ways. However, I found a number of significant flaws in experiments, analytical theory, and modelling, which would make the finding of this work very questionable. I am afraid that I need to reject the publication in Nature Communications. At the same time, I would provide the following comments/suggestions that I hope to be helpful.

[Response] We greatly appreciate the reviewer for her/his taking time to review our paper and her/his evaluation of our work. In our revised manuscript, we have performed more reliable experiments and calculations to support our statements. The following is a point-by-point response to the specific comments.

[Comment] 1. When talking about twisted bilayers, it is really important to assure samples are clean. However, what I found particularly in Supplementary Figure 2 is that the interlayer interfaces have been contaminated seriously – full of wrinkles, bubbles, and so on. With such contamination, it is not fair to claim what has been measured is truly related to the twist-angle-related interface! Better to prepare samples with clear moiré signals.

[Response] We thank the reviewer for the valuable comment. We agree that the bubbles and wrinkles existing at the twisted bilayer interfaces would lead to unpredictable influence on the interlayer interactions. Therefore, cleaner interfaces are more preferred.

In the past several months we have made many efforts to minimize the density of bubbles and wrinkles by keeping our transfer processes clean in all aspects. The details are described in the Methods section of the revised manuscript (page 18 and 19). In our optimized transfer method, we used high-quality and clean MoS₂ triangular single-crystal monolayers (Fig. R1a). Our transfer processes still include a PMMA-assistant wet transfer to break the bottom MoS₂ monolayer region over holes and a PDMS-assistant dry transfer to stack and suspend the upper MoS₂ monolayer region over holes, as shown in Fig. 1a. This random stacking is very efficient for preparing twisted MoS₂ bilayers with various twist angles. We also tried to reduce the data dispersion by further using holey substrates with larger holes (~1.5 μm in diameter) in the nanoindentation to reduce the relative offset of indentation positions and the deviation of measured hole sizes.

Fig. R1 | As-grown (a) and transferred (b, c) MoS₂ triangular monolayers. Clean surfaces and few bubbles/wrinkles can be observed on MoS₂ monolayers after our optimized PMMA-assistant wet transfer (b) and optimized PDMS-assistant dry transfer (c).

Specifically, in the wet transfer, we kept each solution process clean and annealed the bottom MoS₂ monolayers at 350 °C in vacuum ($\sim 1 \times 10^{-3}$ Pa) to remove any PMMA residues. As shown in Fig. R1b, a MoS₂ monolayer transferred onto a flat SiO₂/Si substrate after this step show a very clean surface that is similar to that of the SiO₂/Si substrate. In the dry transfer, PDMS was cured on an atomically flat Si wafer, which avoids the large surface roughness of PDMS that may induce wrinkles of MoS₂ monolayer. Then the dry transfer of the upper MoS₂ monolayer was done in an Ar glove box to avoid interface contaminations from air during the twisted stacking process. At last, the sample was annealed again to remove PDMS residues on the surface and have the twisted bilayer interact effectively. Figure R1c shows a MoS₂ monolayer dry transferred onto a holey SiO₂/Si substrate, which also has a clean surface with few bubbles and wrinkles.

With these efforts, we have effectively reduced the density of bubbles and wrinkles in the twisted region and obtained clean samples on holey substrates (Fig. R2a-c). Compared with the large quantity of bubbles and wrinkles shown in our original manuscript (Fig. R2d and e), the bubbles and wrinkles have been greatly eliminated in the twisted region after our optimized clean transfer processes (Fig. R2b and c).

Fig. R2 | Twisted MoS₂ bilayers on holey substrates. (a) Optical image of a PDMS-transferred upper MoS₂ monolayer (yellow dashed area) overlapping a PMMA-transferred bottom MoS₂ monolayer (white dashed area), forming a twisted region. (b, c) AFM images of a twisted MoS₂ region on a holey substrate made by the optimized transfer processes. (d, e) AFM images of a twisted MoS₂ region made by the previous transfer processes in our original manuscript.

Despite the achievement of clean samples, we note that a thorough elimination of bubbles and wrinkles in the twisted region of 2D materials seems impossible, because in this region the 100% clean interface cannot be obtained. Figure R3 shows a comparison of bubbles coverage ratio between our twisted samples and other reported results. The overall coverage ratio of bubbles and wrinkles in our sample has been close to the lows in the twisted samples reported in literature (e.g., for thick samples, *Nat. Nanotech.* 2021, **16**, 888, *Nat. Comm.* 2019, **10**, 2302; for monolayer samples, *Nat. Comm.* 2020, **11**, 2153). Considering that the materials we twisted are monolayers and the

target substrate is holey, both of which are much easier to cause bubbles or wrinkles than thick flakes or flat substrates, we believe that our twisted samples have already been of very high quality. Clear Moiré patterns observed under annular dark-field scanning transmission electron microscopy (ADF-STEM) also suggest the high quality and clean interfaces of our twisted samples (Fig. R4).

Fig. R3 | Comparison of our twisted samples on holey substrates with others on flat substrates reported in literature. A very low density of bubbles and wrinkles is observed in the twisted region of our monolayer samples on a holey substrate. The bubbles coverage ratio of our samples has reached the lows as reported on flat substrates.

Fig. R4 | Clear Moiré patterns observed under HADF-STEM in our twisted bilayer MoS₂ samples with twist angles of 24° (a) and 9° (b). Insets show the corresponding simulated Moiré patterns, well consistent with the experimental observations.

We also note that in our nanoindentation experiment, the suspended upper MoS₂ monolayer is clamped somewhere around the a hole edge according to the fixed-boundary model or shearing-boundary model. Therefore, the measured moduli would strongly depend on the tightness of the twisted region around the hole edge. This region relatively lacks bubbles and wrinkles because they may merge and move inside the hole. As shown in the most left panel of Fig. R3, the coverage ratio

of bubbles and wrinkles in the 200-nm-width rim of a hole edge is only $\sim 10\%$, reaching to the lowest level reported on flat substrates. With the improved sample quality, our new samples exhibit a modulus of ~ 165 N/m for $\text{MoS}_2@/\text{SiO}_2$ and moduli ranging from 132 to 144 N/m for $\text{MoS}_2@/\text{MoS}_2$ (Fig. R5, red data). The difference in the moduli of $\text{MoS}_2@/\text{SiO}_2$ and $\text{MoS}_2@/\text{MoS}_2$ is $\sim 20\%$, apparently larger than the error possibly induced by the bubbles coverage ($\sim 10\%$), suggesting that our new results are more solid for the statement of the shearing-boundary model. Note that the new measured moduli of $\text{MoS}_2@/\text{SiO}_2$ and $\text{MoS}_2@/\text{MoS}_2$ are very close to our previous results although the new data have much smaller deviation (Fig. R5), which indicates the reliability of our previous results and validates our statement that only a small twisted area around the hole edge affects the moduli measurements.

Fig. R5 | Measured moduli. The $\text{MoS}_2@/\text{MoS}_2$ moduli obtained from 1- μm -diameter holes are previous results (blue circles and cyan stars), while the moduli from 1.5- μm -diameter holes are new results for the samples prepared by the optimized transfer processes (red circles and orange stars).

In short summary, we have greatly improved our transfer processes to reduce the contamination and minimize the density of bubbles and wrinkles. The new data are consistent with our previous results and support the shearing-boundary model in twisted MoS_2 bilayers.

In response to this comment, we added Fig. R1 as Supplementary Fig. 1, Fig. R2a and R2c as Fig. 1b and 1c, Fig. R3 as Supplementary Fig. 3, Fig. R4 as Fig. 1d, and Fig. R5 as Fig. 2c. We also supplemented the above discussion on page 6, 7, and 10-12 of the revised manuscript, and described the preparation of the twisted samples in detail on page 19 and 20 in Methods section.

[Comment] 2. Using indentation to extract stiffness of 2D elastic sheets is not a reliable way though this erroneous method has been propagating in the past decade. First of all, the pretension term in eq (1) is wrong – the asymptotic behaviour suggests a logarithmic dependence on the tip radius so that smaller radii turn out to make a larger contribution (see Vella & Davidovitch 2017). Second, the error produced by the sum of pretension and membrane tension could be on the order of 1 – this ad hoc sum can not be used at all.

[Response] We thank the reviewer for her/his comments and pointing out an important literature. However, we respectfully disagree with the reviewer that using nanoindentation and Eq. (1) to

extract stiffness of 2D elastic sheets is not reliable and even erroneous. Eq. (1) describes the force-displacement F - δ relationship for nanoindentation test on a thin circular sheet under a point force, and includes the asymptotic solutions at the small and large displacement. In Eq. (1), the one-order term related to pretension is dominant at the small displacement, while the cubic term related to in-plane modulus is dominant at the large displacement. As mentioned by the reviewer and the literature (*Soft Mater*, 2017, 13, 226) the reviewer pointed out, Eq. (1) does not consider the influence of indenter tip radius, and even some fitted experimental data in previous studies are not in the range of large displacement. These might induce a certain error. We fully understood some criticisms on inherent limitations of Eq. (1). However, to the best of our knowledge, so far there is no exact analytical and explicit solution to nanoindentation of thin membrane with the clamped boundary due to the inherent geometric nonlinearity. Although Eq. (1) is an approximate formula, it captures the main deformation features (especially, cubic term at large displacement) of nanoindentation and has a simple and explicit expression, hence it has been widely used to extract the in-plane stiffness (or in-plane modulus) of many 2D materials from nanoindentation force-displacement curves. Nearly all measurements on in-plane moduli of 2D materials based on such method are consistent with corresponding results from DFT calculations (graphene: *Science*, 2008, 321, 385; MoS₂: *ACS Nano*, 2011, 5, 9703; WS₂: *Nano Lett*, 2014, 14, 5097; hBN: *Nature Comm*, 2017, 8, 15815). In our current study, we used Eq. (1) to fit the nanoindentation force-displacement curves and to further extract the in-plane modulus of MoS₂ monolayer on MoS₂ substrate. In fact, the accuracy of extracting in-plane modulus from F - δ curve is mainly determined by the cubic term in Eq. (1). If only there are enough experimental data falling in the large displacement regime (*i.e.*, following the cubic term), it is possible to use Eq. (1) to determine the modulus with high precision (*Sci Adv*, 2018, 4, eaat0491). As shown in Fig. 2b, the fitting curve based on Eq. (1) nearly coincides with the experimental curve. The extracted moduli of MoS₂ monolayer from nanoindentation on 1- μ m- and 1.5- μ m-diameter holes are nearly identical and close to previous experimental results. Furthermore, we have exhibited the F - δ curve in logarithmic scale, and it fits well with two asymptotic terms in Eq. (1). For the large displacement, the contribution of pretension is much smaller than that of in-plane stretching related to the in-plane modulus. Thus, we are confident that our results about in-plane modulus of MoS₂ monolayer from nanoindentation are reliable and valid.

In response to the reviewer comment, we have added the following sentences on page 9 and cited the relevant literature (including the reference the reviewer pointed out) to address the main feature, limitation, and application of Eq. (1),

“Eq. (1) includes the asymptotic solutions at small and large displacements. For the small displacement, the one-order term related to pretension is dominant. For the large displacement, the cubic term related to the in-plane modulus is dominant. Because Eq. (1) captures the main deformation features (especially the cubic term at the large displacement) of nanoindentation and has a simple and explicit expression, it has been widely used to extract the in-plane stiffness (or in-plane modulus) of various 2D materials from nanoindentation force-displacement curves^{23,25,26,28,37}. Note that Eq. (1) does not consider the influence of the indenter tip radius, and thus it introduces a certain error³⁸. However, the accuracy of extracting the in-plane modulus from the nanoindentation force-displacement curve is mainly determined by the cubic term in Eq. (1). If only there are enough experimental data falling in the large displacement regime (*i.e.*, following the cubic term), it is possible to use Eq. (1) to determine the modulus with high precision²⁸.”

[Comment] 3. The authors discussed two hypotheses for their finding on the angle-independent moduli. Again, I can not be convinced by the finding itself given the contaminated samples and erroneous fitting method used. Besides, both hypotheses have been explained well in some recent works, in particular Davidovitch & Guinea 2021 and Dai and Lu 2021. What appeared more important is the presence of wrinkling due to the sliding of the vdW interface, as observed in Pablo Ares et al PNAS 2021. The wrinkling would further modify the mechanical response to a point load in a significant manner but is not considered here. Indeed, it is a mess/impossible to consider those important things in one simple expression, which makes the indentation methodology less useful.

[Response] We thank the reviewer for this comment and the references the reviewer provided. As we have noted in the response to the reviewer's comment 1, we have prepared much cleaner twisted samples with clear Moiré patterns. The new measured moduli of MoS₂@SiO₂ and MoS₂@MoS₂ have much smaller statistical deviations and their values are very close to our previous results (Fig. R4), which verifies the reliability of our previous results. We have also argued with the reviewer that the nanoindentation is not an erroneous fitting method in the response to her/his comment 2.

Although Davidovitch & Guinea discussed a theoretical model based on the sliding boundary condition in nanoindentation (*Phys. Rev. E*, 2021, 103, 043002), they did not give an analytic solution. Dai and Lu theoretically presented an implicit solution, rather than an explicit analytic solution, to this problem (*J. Mech. Phys. Solids*, 2021, 149, 104320). In contrast, we show an explicit solution to the shearing-boundary condition in nanoindentation, and more importantly, this solution is supported by our experimental results. Therefore, we believe that our work steps more forward than these previous works.

Regarding the possible wrinkling due to the sliding of a vdW interface, we would like to note that in our experiments, in order to avoid the unrecoverable deformation of samples (e. g. plastic deformation or fracture), we performed the nanoindentation under moderate loads (90-450 nN). In this case, the corresponding indentation depth is much smaller than the diameter of holes (30-70 nm for holes in 1 μm diameter and 60-130 nm for holes in 1.5 μm diameter), and as a result, the strain applied on the upper MoS₂ monolayer is estimated to be less than 2% for all of the nanoindentation measurements. Under such small strains, both the deformation of the suspended upper monolayer region and the shearing at the twisted bilayer region are elastic rather than plastic. As a result, we did not observe any wrinkling before and after nanoindentation (Fig. R10). This fact excludes the wrinkling effect (Ares, et al. *PNAS*, 2021, 118, e2025870118) that may be induced by nanoindentation and simplifies our model. Our work not only verifies the existence of structural softening (Davidovitch & Guinea, *Phys. Rev. E*, 2021, 103, 043002; Dai and Lu, *J. Mech. Phys. Solids*, 2021, 149, 104320), but also put forward an analytic approximate mechanical model to quantitatively study the interlayer shearing.

Fig. R10 | AFM image of a suspended upper MoS₂ monolayer before (a) and after (b) nanoindentation.

In response to this comment, we added Fig. R10 as Supplementary Fig. 8, and the above discussions on page 10 of the revised manuscript. We also cited the references pointed out by the reviewer in the revised manuscript.

[Comment] 4. Eq (2) proposed by the authors is wrong as well in several aspects. First, the pretension term is incorrect as mentioned above; Second, the superposition of three terms for a nonlinear problem is wrong in itself. Three, the deriving of the third used a wrong boundary condition (supplementary eq 8 and 9 where the infinity is subject to a pretension instead of null). Four, thin sheets like MoS₂ cannot sustain any compressive force so the formation of instabilities makes the analysis inside and outside the hole inappropriate.

[Response] We thank the reviewer for her/his comments on Eq. (2) we derived. However, we respectfully disagree with the reviewer that Eq. (2) is wrong. From the scientific point of view, nearly all theoretical models are developed to explain the experimental results, to advance the fundamental understanding for internal relationships among physical variables, and to predict more new results. We thought that the correctness and validity of theoretical model are associated with the facts whether the model follows the basic theories and principles and whether the model can explain the relevant experimental results and even predict more new results.

During the derivation of Eq. (2), we used the basic method in the framework of elastic mechanics theory. When the shear stress τ in Eq. (2) tends to infinity, Eq. (2) can be transformed into Eq. (1), i.e. our model can be degraded to previous fixed-boundary model. Furthermore, we used our model to fit all available force-displacement curves from our experiments. The fitted curves agree well with the experimental curves, as exemplified by Fig. 3e. All the fitted parameters obtained fall in the reasonable range and are consistent with some results reported in previous studies. To further validate our model, we used our model to fit the nanoindentation force-displacement curves of graphene on SiO₂ reported in ref [37] (*Science*, 2008, 321, 385). The obtained values of τ , σ_0 , and E for 1.5 μm -size hole nanoindentation are 3.2 MPa, 810.6 MPa, and 1127.3 GPa, respectively. The obtained values of τ , σ_0 , and E for 1 μm -size hole nanoindentation are 6.8 MPa, 1061.4 MPa, and 1183.6 GPa, respectively. These values of σ_0 and E are close to those reported in ref [37] (*Science*, 2008, 321, 385). Especially, the value of shear stress τ for graphene on SiO₂ substrate from 1.5- μm -diameter hole nanoindentation is close to that (1-3 MPa) measured by pressurized microscale

bubbling (*Phys Rev Lett*, 2017, 119, 036101). All these evidences indicate the validity of our model. We are confident that our model can be used to extract the in-plane modulus of indented membrane and the interlayer shear stress between a membrane and a substrate.

Subsequently, we made the point-by-point responses to argue the correctness and validity of our model (or Eq. (2) we derived).

(1) Pretension term

As mentioned in the responses to the comment 2, at the large displacement, the cubic term is dominant, and the one-order term related to pretention is negligible compared with the cubic term. The accuracy of extracting in-plane modulus from F - δ curve is mainly determined by the cubic term in Eq. (1). If only there are enough experimental data falling in the large displacement regime (i.e. following the cubic term), it is possible to use Eq. (1) to determine the modulus with high precision. The same situation is also for Eq. (2).

(2) Superposition of three terms

The superposition of the first and second terms in Eq. (2) is the same as that in Eq. (1). The presence of the third term in Eq. (2) is not a simple superposition, but from an analytic derivation in consideration of introducing a shear zone. During our derivation, the membrane is divided into two parts: one is the membrane (under plane-stress state) subjected to interfacial shear stress outside the hole ($r > a$, a is the radius of the hole), and the other is the membrane subjected to applied point load in suspended region ($r < a$). It is noted that two parts should satisfy the continuity conditions of displacement and stress in the radial direction at the edge of the hole ($r = a$). The third term is introduced to satisfy the continuity conditions in terms of radial stress and displacement at the edge of the hole ($r = a$). When the shear stress τ in Eq. (2) tends to infinity, the third term in Eq. (2) tends to be zero, and Eq. (2) can be transformed into Eq. (1), i.e. our model can be degraded to previous fixed-boundary model. It indicates that the third term in Eq. (2) originates from the interlayer shear stress between the membrane and the substrate.

(3) Boundary condition for deriving the third term in Eq. (2)

As mentioned in the above responses, the membrane in our model is divided into two parts: one is the membrane (under plane-stress state) subjected to interfacial shear stress outside the hole ($r > a$, a is the radius of the hole), and the other is the membrane subjected to applied point load in suspended region ($r < a$). The third term in Eq. (2) is introduced to satisfy the continuity conditions in terms of radial stress and displacement at the edge of the hole ($r = a$). For the membrane outside the hole, Eqs. (8) and (9) in Supplementary Information represent the radial and circumferential resultant stresses due to the presence of shear stress, respectively. When considering the continuity condition of radial stress at $r = a$ (i.e., Eq. (17) in Supplementary Information), the contribution of pretension is indeed introduced in the radial stress of membrane inside the hole (see the first term on the left side of Eq. (17)).

(4) Compressive force

During our derivation, the introduction of homogeneous stress σ_c independent of r and shear zone size ρ are used to establish the continuity conditions of radial displacement and stress at the edge of hole (see Eqs. (16) and (17) in Supplementary Information). When the model is initially established, both σ_c and ρ are unknown. Their expressions are determined by the continuity conditions (see Eqs. (18) and (19) in Supplementary Information). Although σ_c is a compressive stress, the overall

membrane is subjected to a summation of pretension σ_0 , homogeneous stress σ_c , and applied stress. The net of these three stresses is the tensile stress.

Based on the above arguments, we are confident that our model is valid and correct. Eq. (2) is derived on the basis of Eq. (1), and therefore Eq. (2) is an approximate formula. However, Eq. (2) reflects the influence of interlayer shear on nanoindentation and has a simple and explicit expression. We are confident that Eq. (2) can be used to extract the in-plane modulus of indented membrane and the interlayer shear stress between a membrane and a substrate.

In response to the comments, we have added some parts of above discussions as the part 7 in the revised Supplementary Information to address the validity of our model.

[Comment] 5. In MD simulations, LJ potential used here is not a good choice (even erroneous) to elucidate the angle-dependent interactions. KC is way better. To better match what is going on in experiments, an axisymmetric model is recommended, which may be used to verify equation (2) since it is the key to the measurement.

[Response] We thank the reviewer for her/his comments on our MD simulations. In our MD simulations, the LJ potential is used to describe the long-range interaction between the membrane and the substrate. In recent years, with the development of van der Waals heterostructures, the LJ potential has been widely used to investigate the dependences of interlayer energy and friction on twisting angle in various van der Waals heterostructures (*Nat. Mater.*, 2022, 21, 47; *Trib. Inter.*, 2020, 151, 106483; *ACS Omega*, 2020, 5, 31692). The LJ potential allows us to treat a simulated system containing more than 1 million atoms, which is far beyond the length scale of DFT calculations.

In the current study, we performed large-scale MD simulations for nanoindentation of MoS₂ monolayer on MoS₂ and SiO₂ substrates to complement the experimental results. The set-up of MD simulations is very similar to the experimental system (Fig. 4b). We simulated the MoS₂ monolayer with different twisting angles with respect to the MoS₂ substrate (Fig. 4c). Figure 4d shows the typical nanoindentation curves from our MD simulations. In the large displacement regime, the scaling exponents of nanoindentation force with respect to displacement are about 2.50 for MoS₂@MoS₂ and 2.74 for MoS₂@SiO₂, which are close to those of experimental curves shown in Fig. 3a and Supplementary Fig. 5b. Such nonlinear behavior is attributed to the common contributions of membrane stretching related to the elastic modulus and the interfacial shear between the membrane and the substrate. Our simulation results also showed that for different twisting angles, the nanoindentation curves nearly coincide with each other. This result is consistent with the experimental results shown in Supplementary Fig. 9.

Furthermore, we performed MD simulations to estimate the shear stress for MoS₂@MoS₂ and MoS₂@SiO₂. Our MD simulations showed that the average shear stresses of MoS₂@MoS₂ and MoS₂@SiO₂ are up to 4.08 MPa and 13.69 MPa, respectively. These values are close to those (2.68 MPa and 11.80 MPa) from our theoretical fitting. To further verify the MD simulation results, we also performed the DFT calculations to characterize the interlayer shear stress for MoS₂@MoS₂. The average interlayer shear stress (4.87 MPa) along the minimum energy path from our DFT calculations is close to that (4.08 MPa) from our MD simulations. These results indicate that our MD simulations with LJ potential are reliable and that relevant simulation results provide atomistic and mechanistic insights and fundamental understanding for the current experimental results.

Moreover, our current MD simulation is fully three-dimensional, which contains more

information and provides more details than an axisymmetric simulation. At the same time, it is difficult to simulate an axisymmetric system in atomistic simulations, since it needs to impose more constraints on the simulated system to satisfy the axisymmetric condition.

REVIEWER COMMENTS

Reviewer #1 (Remarks to the Author):

They authors addressed all my comments and suggestions, hence I recommend the manuscript for publication in Nature Communications.

Reviewer #2 (Remarks to the Author):

The author has clarified all my concerns. In my opinion, the paper has considerably improved after these important changes and is now suitable for publication.

Reviewer #3 (Remarks to the Author):

I appreciate that the authors have made extensive efforts to address the concerns I had when reviewing the earlier version of the manuscript. For papers with an emphasis on characterization, I typically have two requirements: the experiments are as accurate as possible and the theory for interpreting measured data is as accurate as possible. I partially understand that the authors have pushed the limit to prepare clean samples for measures during the revision. Now I also understand that the authors had chosen to use simplified analytical models such that the data is easily interpreted.

Merely illustrating the authors' interpretation of this problem, the use of such models may be acceptable but the errors due to these models need to be clearly stated and explained. First-principles modeling with more formal analysis could be pursued in the future by the authors or by others. The authors had clarified the errors intrinsically associated with eqn (1) in the revision, which I am content with now.

However, still a concern I hope the authors address before recommending the publication is to clearly state in the main text the errors caused when using eqn (2) to fit out the shear stress. This is important because 1) future follow-up experimental works may have an understanding of the source of errors if they want to adopt or have adopted the method proposed in this work and 2) future careful theoretical works may have a focus on resolving these errors though they may turn out too complex to have an analytical form.

Sorry for being harsh but again these comments are all about two basic requirements for a good paper aiming at accurate characterizations.

Reviewer #1 (Remarks to the Author):

[Comment] They authors addressed all my comments and suggestions, hence I recommend the manuscript for publication in Nature Communications.

[Response] We thank the reviewer for her/his taking time to review our manuscript again and the recommendation of publication of our manuscript.

Reviewer #2 (Remarks to the Author):

[Comment] The author has clarified all my concerns. In my opinion, the paper has considerably improved after these important changes and is now suitable for publication.

[Response] We thank the reviewer for her/his taking time to review our manuscript again and the recommendation of publication of our manuscript.

Reviewer #3 (Remarks to the Author):

[Comment] I appreciate that the authors have made extensive efforts to address the concerns I had when reviewing the earlier version of the manuscript. For papers with an emphasis on characterization, I typically have two requirements: the experiments are as accurate as possible and the theory for interpreting measured data is as accurate as possible. I partially understand that the authors have pushed the limit to prepare clean samples for measures during the revision. Now I also understand that the authors had chosen to use simplified analytical models such that the data is easily interpreted.

[Response] We greatly appreciate the reviewer's comments and suggestions on our revision. All of the comments/suggestions are considered to be very helpful for improving our manuscript.

[Comment] Merely illustrating the authors' interpretation of this problem, the use of such models may be acceptable but the errors due to these models need to be clearly stated and explained. First-principles modeling with more formal analysis could be pursued in the future by the authors or by others. The authors had clarified the errors intrinsically associated with eqn (1) in the revision, which I am content with now.

[Response] We thank the reviewer for the positive comments on our discussion/clarification of errors associated with Eq. (1) and the suggestions about the first-principles modeling. In the last revision, the first-principles modeling was used to verify and complement the results about the interlayer shear stresses obtained from molecular dynamics simulations. As the reviewer mentioned, more detailed analyses for first-principles modeling and even more accurate and reliable modeling will be performed in the future by the researchers in the community of low-dimensional materials.

[Comment] However, still a concern I hope the authors address before recommending the publication is to clearly state in the main text the errors caused when using eqn (2) to fit out the shear stress. This is important because 1) future follow-up experimental works may have an understanding of the source of errors if they want to adopt or have adopted the method proposed in this work and 2) future careful theoretical works may have a focus on resolving these errors though they may turn out too complex to have an analytical form.

[Response] We thank the reviewer for the helpful suggestion. It is very challenging and even impossible to quantitatively analyze the errors caused when using our theoretical model (Eq. (2)) to fit the interlayer shear stress, because there has been no exact solution to nanoindentation on the elastic membrane which suspends on a hole and has a shearing-boundary condition. We have ever tried to use the finite-element modelling to mimic such nanoindentation with the same conditions as the real experiments. However, it is very difficult to describe the nonlinear shear stress distribution between suspended membrane and substrate, because such shear stress distribution in the real experiments is actually unknown.

Previous theoretical study (J. Mech. Phys. Solids, 2021, 149, 104320, i.e., Ref. 41 in the manuscript) gave an implicit expression for nanoindentation on the suspended elastic membrane. This study considered the shear between membrane and substrate and the resultant wrinkling, but excluded the pretension on the membrane. Except pretension, most conditions involved in this study are very similar to those in the real experiments. To estimate the errors, we performed numerical calculations based on the implicit expression in this study and then obtained the indentation force-displacement data. We validated that the obtained data are completely the same as those shown in this study. We further used the Eq. (2) to fit these data and obtained the corresponding interlayer shear stress τ and pretension σ_0 , when the modulus of membrane (i.e., MoS₂ single layer) is given. Figure R1 shows the comparison of our fitting force-displacement curves with the numerical solutions to the implicit expression. As shown in Fig. R1, the fitting curves based on the Eq. (2) nearly coincide with the numerical solutions to the implicit expression, and both interlayer shear stress and pretension from the fitting are comparable or close to original values in the implicit solution. The errors of interlayer shear stresses are attributed to the approximation and simplification we used during the derivation of Eq. (2).

Figure R1. Comparison of our fitting force-displacement curves with the numerical solutions to the implicit expression (J. Mech. Phys. Solids, 2021, 149, 104320, i.e., Ref. 41 in the manuscript).

In the light of the reviewer's suggestion, we added the following statements on page 16 of the revised manuscript to address the error caused when using the Eq. (2) to fit the interlayer shear stress and to further analyze the source of the error,

“However, there exists a certain error induced by using Eq. (2) to fit the experimental results, since Eq. (2) is an approximate solution for indentation of ultrathin elastic membrane with the shearing-boundary condition. The error might mainly originate from the approximation and simplification during the derivation of Eq. (2): (i) ignoring

the finite size of indenter, (ii) simplifying nonlinear distribution of interlayer shear stress between the membrane and substrate, and (iii) simplifying complex coupling/interplay among in-plane stiffness, out-of-plane deflection, pretension, and interlayer shear.”

[Comment] Sorry for being harsh but again these comments are all about two basic requirements for a good paper aiming at accurate characterizations.

[Response] We thank the reviewer very much again for all of comments and suggestions in these two rounds of reviews, which are very helpful and useful for improving our manuscript.

REVIEWERS' COMMENTS

Reviewer #3 (Remarks to the Author):

The revision has partially addressed the concerns and I think this version is publishable.